

# Renormalization approach to the superconducting Kondo model

**Steffen Sykora⋆ and Tobias Meng**

Institute for Theoretical Physics and Würzburg-Dresden Cluster of Excellence ct.qmat,
Technische Universität Dresden, 01069 Dresden, Germany

⋆ steffen.sykora@tu-dresden.de

## Abstract

An approach to bound states based on unitary transformations of Hamiltonians is presented. The method is applied to study the interaction between electrons in a BCS $s$-wave superconductor and a quantum spin. It is shown that known results from the t-matrix method and numerical studies are reproduced by this new method. As a main advantage, the method can straightforwardly be extended to study the topological properties of combined bound states in chains of many magnetic impurities. It also provides a uniform picture of the interplay between the Yu-Shiba-Rusinov (YSR) bound states and the Kondo singlet state.



# 1 Introduction

The study of individual impurity bound states in electronic systems has become an important experimental and theoretical tool for the characterization of correlated quantum materials. A prominent example is a system in which a magnetic impurity interacts with a superconducting condensate, leading to the formation of Yu-Shiba-Rusinov (YSR) bound states [1–3], but also to Kondo physics [4–6]. Depending on the strength of the exchange coupling, a transition from a regime dominated by weakly coupled YSR states to a more Kondo-like state is found [7–9]. This physics has at least in parts be visualized by scanning tunneling spectroscopy experiments [10–12].

In addition, the controlled extension to more than one local moment has recently become an important theme. This gives rise to models of molecules [13–16] and spin chains [17, 18]. The interest in these systems has prominently been fueled by predictions of topological superconductivity and Majorana zero modes [19, 20]. In the discussion of this intriguing physics, however, the influence of the quantum nature of the impurity spins, and the resulting Kondo physics has not been explored extensively. This is in parts due to the fact that the corresponding theoretical treatment requires true many-body approaches that properly take into account the quantum nature of an entire collection of local moments.

In recent years, some powerful theoretical methods like mean field calculations [7, 21], perturbation theory [22, 23], self-consistent approaches [24–26] and the numerical renormalization group (NRG) approach [27–29] have been applied to study the formation of bound states in correlated systems. Many of these methods are restricted to only a very small number of quantum impurities, or to very small system sizes. In particular, powerful analytical approaches which are able to tackle a collection of quantum impurities in an environment of a condensate in the thermodynamic limit are not available so far. Such an approach would allow for a deeper understanding of the underlying physical processes and the structure of the quasiparticle which is responsible for the bound state. In addition, the quantum nature of the local impurity is often important for the properties of the spectral function.

Motivated by these open questions, we have developed a new renormalization scheme for Hamiltonians describing a local moment interacting with a superconducting condensate. Our diagonalization method shares some basic concepts with the known flow equation approaches [30–32]. The method is technically organized in such a way that the formation of bound states is highlighted. A schematic picture of the method is shown in Fig. 1. As we discuss below, bound states lead to singularities in the unitary transformation on which our method builds. These singularities in turn lead to additional local contributions to the quasiparticle operators (the bound state), and to a significant energy renormalization (the bound state energy). The quantum nature of the impurity spin is taken into account on an approximate level, but we show that this approximation is sufficiently refined to allow access to Kondo physics. Due to the readily generalizable unitary transformations used to diagonalize the Hamiltonian, also more than one impurity can in principle be implemented (for normal state electronic Kondo lattices, this has already been worked out in Ref. [33]).

The rest of the paper is organized as follows. In Sec. 2, we introduce the model Hamiltonian. Sec. 2.1 is devoted to a detailed discussion of our renormalization method. In particular, we explain the method of integrating out the electron-impurity spin interaction, and show how the corresponding renormalization equations are derived. Then, in Sec. 2.2, we explain the appearance of bound states in our theory, and how the bound states are related to singularities in the unitary transformation. In Sec. 2.3, we discuss the access to expectation values within our approach. We apply the developed concepts to renormalize single-particle operators and present two examples of expectation values, the occupation number and the single-particle spectral function. Sec. 3 is devoted to numerical results for the YSR bound state energy and

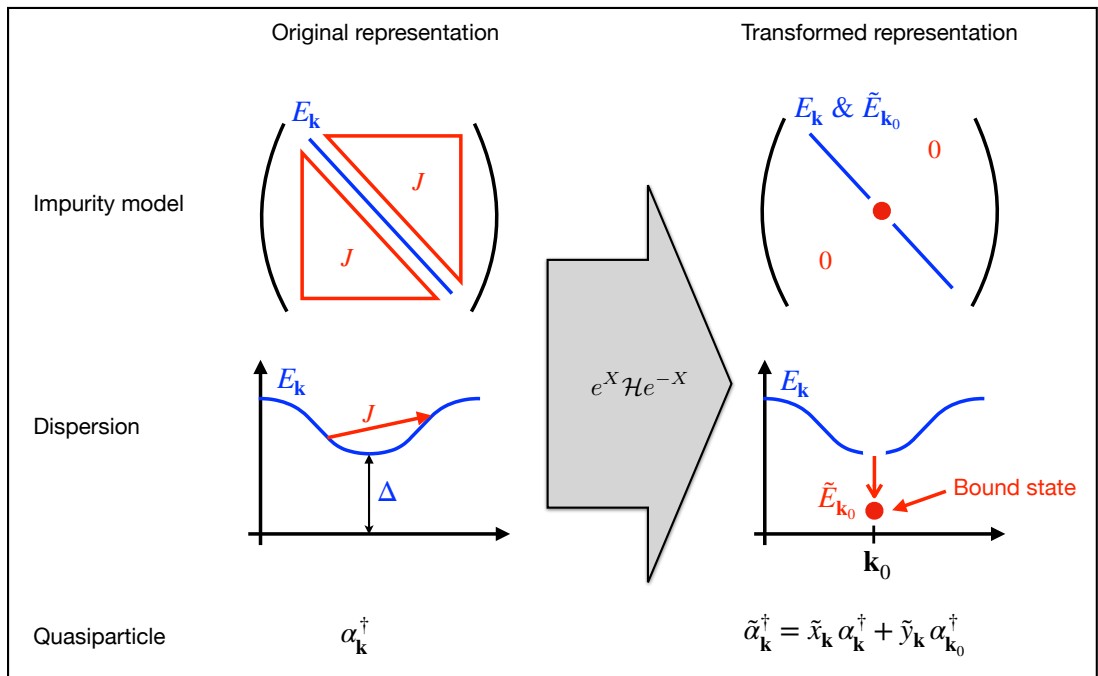

Figure 1: Schematic picture of our theoretical approach to bound states. The scattering term $\sim J$ in the original Hamiltonian is eliminated by a unitary transformation $e^X$. This technique diagonalizes the Hamiltonian and generates a bound state with quantum number $\mathbf{k}_0$ (red dot), which is renormalized in energy. Excitations split into a coherent part and a local excitation.

the single-particle spectral function in the weak and strong coupling regimes, which we relate to the nature of the excitations to highlight the coexistence and competition of YSR and Kondo physics. We furthermore show that our methods compared very well with other methods, in particular by benchmarking with numerical renormalization group (NRG) studies. Finally, a conclusion is given in Sec. 4.

## 2 Theoretical approach

Our starting point is a basic BCS-type $s$-wave superconductor coupled to a single magnetic impurity. Denoting the spin by an index $\sigma = \uparrow, \downarrow$, this system is modelled by the Hamiltonian

$$\mathcal{H} = \mathcal{H}_0 + \mathcal{H}_1\,, \quad \text{with} \quad \mathcal{H}_0 = \sum_{\mathbf{k}\sigma} \varepsilon_{\mathbf{k}} c^{\dagger}_{\mathbf{k}\sigma} c_{\mathbf{k}\sigma} + \Delta \sum_{\mathbf{k}} (c^{\dagger}_{\mathbf{k}\uparrow} c^{\dagger}_{-\mathbf{k}\downarrow} + c_{-\mathbf{k}\downarrow} c_{\mathbf{k}\uparrow}) \quad \text{and} \quad \mathcal{H}_1 = J\mathbf{S} \cdot \mathbf{s}_{\mathbf{r}_{\text{imp}}}\,,$$

(1)

where $\mathbf{S}$ and $\mathbf{s}_{\mathbf{r}_{\text{imp}}}$ are the local impurity spin and the electron spin at the impurity site $\mathbf{r}_{\text{imp}}$, respectively. Here, $\varepsilon_{\mathbf{k}}$ is the bare electronic dispersion, which is a function of the momentum vector $\mathbf{k}$, and $\Delta$ is the superconducting gap. The lattice consists of $N$ sites. We consider the case of a single local magnetic impurity represented by a quantum spin of arbitrary size that is described by an angular momentum operator $\mathbf{S}$. This impurity is coupled to the electrons via a Heisenberg exchange coupling $J$, and the coordinate origin is chosen to coincide with the impurity position.

Before dealing with the magnetic impurity, our approach requires to diagonalize the purely electronic Hamiltonian $\mathcal{H}_0$. This is achieved by introducing the usual Bogoliubov quasiparti-

cles,

$$\alpha_{\mathbf{k}}^{\dagger} = u_{\mathbf{k}} c_{\mathbf{k}\uparrow}^{\dagger} - v_{\mathbf{k}} c_{-\mathbf{k}\downarrow}, \quad \beta_{\mathbf{k}}^{\dagger} = u_{\mathbf{k}} c_{-\mathbf{k}\downarrow}^{\dagger} + v_{\mathbf{k}} c_{\mathbf{k}\uparrow}, \tag{2}$$

with

$$u_{\mathbf{k}}^{2} = \frac{1}{2}\left(1 + \frac{\varepsilon_{\mathbf{k}}}{\sqrt{\varepsilon_{\mathbf{k}}^{2} + \Delta^{2}}}\right), \quad v_{\mathbf{k}}^{2} = \frac{1}{2}\left(1 - \frac{\varepsilon_{\mathbf{k}}}{\sqrt{\varepsilon_{\mathbf{k}}^{2} + \Delta^{2}}}\right). \tag{3}$$

The electronic Hamiltonian $\mathcal{H}_0$ is thus brought to the diagonal form

$$\mathcal{H}_0 = \sum_{\mathbf{k}} E_{\mathbf{k}}\left(\alpha_{\mathbf{k}}^{\dagger}\alpha_{\mathbf{k}} + \beta_{\mathbf{k}}^{\dagger}\beta_{\mathbf{k}}\right) + \sum_{\mathbf{k}}\left(\varepsilon_{\mathbf{k}} - E_{\mathbf{k}}\right), \tag{4}$$

with the quasiparticle energy $E_{\mathbf{k}} = \sqrt{\varepsilon_{\mathbf{k}}^{2} + \Delta^{2}}$. Next, we turn to the electron-impurity coupling. In the original basis, it reads

$$\mathcal{H}_1 = \frac{J}{N}\sum_{\mathbf{k}\mathbf{k}'}\sum_{\alpha,\beta}\mathbf{S}\cdot\frac{\vec{\sigma}_{\alpha\beta}}{2}c_{\mathbf{k}\alpha}^{\dagger}c_{\mathbf{k}'\beta}. \tag{5}$$

Replacing the conduction electron operators with the ones of Bogoliubov quasiparticles using Eq. (2), and introducing the shorthands

$$C_{1,\mathbf{k}\mathbf{k}'}^{\pm} = u_{\mathbf{k}}u_{\mathbf{k}'} \pm v_{\mathbf{k}}v_{\mathbf{k}'}, \quad C_{2,\mathbf{k}\mathbf{k}'}^{\pm} = u_{\mathbf{k}}v_{\mathbf{k}'} \pm v_{\mathbf{k}}u_{\mathbf{k}'}, \tag{6}$$

for combinations of Bogoliubov coefficients (coherence factors), the electron-impurity-coupling takes the form

$$\begin{aligned}
\mathcal{H}_1 = \frac{J}{2N}\sum_{\mathbf{k}\mathbf{k}'}\Big\{ & C_{1,\mathbf{k}\mathbf{k}'}^{+}\left(\alpha_{\mathbf{k}}^{\dagger}\alpha_{\mathbf{k}'}S_z - \beta_{\mathbf{k}}^{\dagger}\beta_{\mathbf{k}'}S_z + \alpha_{\mathbf{k}}^{\dagger}\beta_{\mathbf{k}'}S^- + \beta_{\mathbf{k}'}^{\dagger}\alpha_{\mathbf{k}}S^+\right) \\
& + C_{2,\mathbf{k}\mathbf{k}'}^{-}\left[\left(\alpha_{\mathbf{k}}^{\dagger}\beta_{\mathbf{k}'}^{\dagger} + \beta_{\mathbf{k}'}\alpha_{\mathbf{k}}\right)S_z + \frac{1}{2}\left(\beta_{\mathbf{k}'}\beta_{\mathbf{k}} - \alpha_{\mathbf{k}}^{\dagger}\alpha_{\mathbf{k}'}^{\dagger}\right)S^- + \frac{1}{2}\left(\beta_{\mathbf{k}}^{\dagger}\beta_{\mathbf{k}'}^{\dagger} - \alpha_{\mathbf{k}'}\alpha_{\mathbf{k}}\right)S^+\right]\Big\}.
\end{aligned} \tag{7}$$

Here, $S^+$ ($S^-$) is the spin raising (lowering) operator, while $S_z$ denotes the $z$-component of the spin operator $\mathbf{S}$. At this point, all we demand of these operators is that they satisfy the commutation relations $[S^+, S^-] = 2S_z$, $[S_z, S^-] = -S^-$, and $[S_z, S^+] = S^+$. In the following, we show that the Hamiltonian $\mathcal{H}_1$ can be diagonalized by a unitary transformation in combination with a factorization scheme. As we discuss, this approach is able to describe the presence of the Yu-Shiba-Rusinov (YSR) bound states [1–3], but also of Kondo physics.

## 2.1 Renormalized Hamiltonian

Unlike the case of a classical spin in a superconductor, or the even simpler case of a potential impurity in a spinless electron bath, the case of a quantum spin immersed into an $s$-wave superconductor constitutes an interacting quantum problem, and therefore cannot be solved exactly. Our approach is built to tackle this interaction on a refined approximate level. The power of our approach can be gauged from the fact that a variant of our approximation scheme has already been used to successfully describe the Kondo effect of a quantum spin coupled to a normal-state electron system [33]. It can therefore be expected to also yield excellent results when applied to a superconducting system. After discussing technical details of the transformation we use to tackle the problem, we will first compare our results with recent experimental and numerical studies, and then show original results for the $\mathbf{k}$-resolved spectral function. We furthermore discuss that our approach clearly illustrates how Kondo physics appears. The latter is associated with an energy scale $k_B T_K$ (with $k_B$ being the Boltzmann constant and $T_K$ the Kondo temperature), and competes with the singlet state of the superconductor [34].

The diagonalization method to be used here is reviewed in Ref. [35]. Following the spirit of this approach, we subject the Hamiltonian (1) to a mapping that brings the Hamiltonian to a diagonal form $\tilde{\mathcal{H}}$, which in turn can be used to calculate physical observables. According to Ref. [35], in a general situation of an interacting many-body problem the diagonalization procedure of the Hamiltonian needs to be carried out in a stepwise way. In the present case, we show that this multiple-step procedure can be constructed such that the coupling term in the Hamiltonian is continuously reduced until it is finally fully eliminated. At each step, the unitary transformation can be limited to the lowest order with respect to a small variable which accounts for the difference between the control parameter values of two subsequent transformation steps. This allows the controlled derivation of renormalization equations for the parameters of the transformed Hamiltonian, which in practice are solved numerically. Eventually, we end up with an effectively free system of Bogoliubov quasiparticles, $\tilde{\mathcal{H}}$, whose renormalized energies account for the effects of the magnetic impurity.

The mapping between the original Hamiltonian $\mathcal{H}$ and an arbitrary intermediate Hamiltonian $\mathcal{H}_\lambda$ appearing during the transformation process is implemented by a unitary transformation,

$$\mathcal{H}_\lambda = e^{X_\lambda}\mathcal{H}e^{-X_\lambda}\,, \tag{8}$$

where the hermiticity of $\mathcal{H}_\lambda$ requires $X_\lambda^\dagger = -X_\lambda$. The dimensionless control parameter $\lambda$ describes the progress of the diagonalization. It is defined such that its initial value $\lambda = 1$ marks the starting point where $\mathcal{H}_{\lambda=1} = \mathcal{H}$, and its final value $\lambda = 0$ describes the fully diagonalized effective Hamiltonian $\mathcal{H}_{\lambda=0} = \tilde{\mathcal{H}}$, with

$$\tilde{\mathcal{H}} = \sum_{\mathbf{k}} \tilde{E}_{\mathbf{k}}\left(\alpha_{\mathbf{k}}^\dagger\alpha_{\mathbf{k}} + \beta_{\mathbf{k}}^\dagger\beta_{\mathbf{k}}\right) + \tilde{E}\,, \tag{9}$$

where finding the renormalized dispersion $\tilde{E}_{\mathbf{k}}$ and the energy constant $\tilde{E}$ is the aim of the diagonalization process. Note that since $\tilde{\mathcal{H}}$ is related to $\mathcal{H}$ by a unitary transformation (and a factorization scheme, see below), the energy values $\tilde{E}_{\mathbf{k}}$ correspond (to a very good approximation) to the eigenvalues of the original Hamiltonian, including the binding energies of possible bound states. Using the introduced definition of $\lambda$, we define the transformed Hamiltonian for an arbitrary $\lambda$ as

$$\begin{aligned}
\mathcal{H}_\lambda =\ & \sum_{\mathbf{k}} E_{\mathbf{k},\lambda}\left(\alpha_{\mathbf{k}}^\dagger\alpha_{\mathbf{k}} + \beta_{\mathbf{k}}^\dagger\beta_{\mathbf{k}}\right) + E_\lambda \\
& + \lambda\frac{J}{2N}\sum_{\mathbf{k}\neq\mathbf{k}'}\Bigg\{C_{1,\mathbf{k}\mathbf{k}'}^+\left(\alpha_{\mathbf{k}}^\dagger\alpha_{\mathbf{k}'}S_z - \beta_{\mathbf{k}}^\dagger\beta_{\mathbf{k}'}S_z + \alpha_{\mathbf{k}}^\dagger\beta_{\mathbf{k}'}S^- + \beta_{\mathbf{k}'}^\dagger\alpha_{\mathbf{k}}S^+\right) \\
& + C_{2,\mathbf{k}\mathbf{k}'}^-\left[\left(\alpha_{\mathbf{k}}^\dagger\beta_{\mathbf{k}'}^\dagger + \beta_{\mathbf{k}'}\alpha_{\mathbf{k}}\right)S_z + \frac{1}{2}\left(\beta_{\mathbf{k}'}\beta_{\mathbf{k}} - \alpha_{\mathbf{k}}^\dagger\alpha_{\mathbf{k}'}^\dagger\right)S^- + \frac{1}{2}\left(\beta_{\mathbf{k}}^\dagger\beta_{\mathbf{k}'}^\dagger - \alpha_{\mathbf{k}'}\alpha_{\mathbf{k}}\right)S^+\right]\Bigg\}.
\end{aligned} \tag{10}$$

Up to the parameter $\lambda$, the second and third lines correspond to the original interaction part $\mathcal{H}_1$.

The renormalization equations for $E_{\mathbf{k},\lambda}$ and $E_\lambda$ are found from a small transformation step which maps $\mathcal{H}_\lambda$ to the corresponding effective Hamiltonian $\mathcal{H}_{\lambda-\Delta\lambda}$ referring to a somewhat smaller parameter $(\lambda - \Delta\lambda)$ with $\Delta\lambda \ll 1$. Thereby we demand that both $\mathcal{H}_\lambda$ and $\mathcal{H}_{\lambda-\Delta\lambda}$ fulfill Eq. (10) and are related to each other via a unitary transformation,

$$\mathcal{H}_{\lambda-\Delta\lambda} = e^{X_{\lambda,\Delta\lambda}}\mathcal{H}_\lambda e^{-X_{\lambda,\Delta\lambda}}\,, \tag{11}$$

where the generator $X_{\lambda,\Delta\lambda}$ has to be constructed such that $\mathcal{H}_{\lambda-\Delta\lambda}$ keeps the structure of Eq. (10) but has a slightly reduced coupling term. It in that sense is closer to a diagonalized Hamiltonian than $\mathcal{H}_\lambda$. For the generator $X_{\lambda,\Delta\lambda}$ satisfying these requirements, we make

the following ansatz

$$
\begin{aligned}
X_{\lambda,\Delta\lambda} = \frac{\Delta\lambda}{N} \sum_{\mathbf{k}\neq\mathbf{k'}} &\Bigg\{ (J_{\mathbf{k},\lambda}+J_{\mathbf{k'},\lambda}) \bigg[ \frac{C^+_{1,\mathbf{kk'}}}{E_{\mathbf{k},\lambda}-E_{\mathbf{k'},\lambda}} \big( \alpha^\dagger_{\mathbf{k}}\alpha_{\mathbf{k'}}S_z - \beta^\dagger_{\mathbf{k}}\beta_{\mathbf{k'}}S_z + \alpha^\dagger_{\mathbf{k}}\beta_{\mathbf{k'}}S^- + \beta^\dagger_{\mathbf{k'}}\alpha_{\mathbf{k}}S^+ \big) \\
&+ \frac{C^-_{2,\mathbf{kk'}}}{E_{\mathbf{k},\lambda}+E_{\mathbf{k'},\lambda}} \Big( (\alpha^\dagger_{\mathbf{k}}\beta^\dagger_{\mathbf{k'}} - \beta_{\mathbf{k'}}\alpha_{\mathbf{k}})S_z - \frac{1}{2}(\beta_{\mathbf{k'}}\beta_{\mathbf{k}} + \alpha^\dagger_{\mathbf{k}}\alpha^\dagger_{\mathbf{k'}})S^- + \frac{1}{2}(\beta^\dagger_{\mathbf{k}}\beta^\dagger_{\mathbf{k'}} + \alpha_{\mathbf{k'}}\alpha_{\mathbf{k}})S^+ \Big) \bigg] \\
&- (V_{\mathbf{k},\lambda}+V_{\mathbf{k'},\lambda}) \bigg[ \frac{C^+_{1,\mathbf{kk'}}}{E_{\mathbf{k},\lambda}-E_{\mathbf{k'},\lambda}} \big( \alpha^\dagger_{\mathbf{k}}\alpha_{\mathbf{k'}} + \beta^\dagger_{\mathbf{k}}\beta_{\mathbf{k'}} \big) - \frac{C^-_{2,\mathbf{kk'}}}{E_{\mathbf{k},\lambda}+E_{\mathbf{k'},\lambda}} \big( \alpha^\dagger_{\mathbf{k}}\beta^\dagger_{\mathbf{k'}} - \beta_{\mathbf{k'}}\alpha_{\mathbf{k}} \big) \bigg] \Bigg\}.
\end{aligned}
\tag{12}
$$

Importantly, this generator is proportional to the small parameter $\Delta\lambda$. It consists of two types of terms. The first two lines contain terms that occur in the original interaction part $\mathcal{H}_1$, and that are composed of products of fermionic and spin operators. The third line, on the contrary, contains only terms independent of the spin and quadratic in the fermionic operators. As we will show below, terms of this form are generated during the diagonalization process, and must therefore also be included in the generator. The unknown coefficients $J_{\mathbf{k},\lambda}$ and $V_{\mathbf{k},\lambda}$, finally, will be determined in such a way that after evaluating the right-hand side of Eq. (11) using the ansatz (12), the form of Eq. (10) is exactly reproduced, except that the parameter $\lambda$ is replaced by $(\lambda-\Delta\lambda)$. The step width $\Delta\lambda \ll 1$ can be chosen arbitrarily small. We can thus approximate the unitary transformation in Eq. (11) by

$$
\mathcal{H}_{\lambda-\Delta\lambda} \approx \mathcal{H}_\lambda + [X_{\lambda,\Delta\lambda}, \mathcal{H}_\lambda].
\tag{13}
$$

In order to find the equations for $J_{\mathbf{k},\lambda}$ and $V_{\mathbf{k},\lambda}$, as well as for $E_{\mathbf{k},\lambda}$ and $E_\lambda$, we have to compute the commutator between $X_{\lambda,\Delta\lambda}$ and $\mathcal{H}_\lambda$. To facilitate the discussion of the various resulting terms, let us first formally decompose the renormalized Hamiltonian into its diagonal and non-diagonal parts, $\mathcal{H}_\lambda = \mathcal{H}_{0,\lambda} + \mathcal{H}_{1,\lambda}$, with

$$
\mathcal{H}_{0,\lambda} = \sum_{\mathbf{k}} E_{\mathbf{k},\lambda} \big( \alpha^\dagger_{\mathbf{k}}\alpha_{\mathbf{k}} + \beta^\dagger_{\mathbf{k}}\beta_{\mathbf{k}} \big) + E_\lambda,
\tag{14}
$$

and

$$
\begin{aligned}
\mathcal{H}_{1,\lambda} = \lambda \frac{J}{2N} \sum_{\mathbf{k}\neq\mathbf{k'}} &\Big\{ C^+_{1,\mathbf{kk'}} \big( \alpha^\dagger_{\mathbf{k}}\alpha_{\mathbf{k'}}S_z - \beta^\dagger_{\mathbf{k}}\beta_{\mathbf{k'}}S_z + \alpha^\dagger_{\mathbf{k}}\beta_{\mathbf{k'}}S^- + \beta^\dagger_{\mathbf{k'}}\alpha_{\mathbf{k}}S^+ \big) \\
&+ C^-_{2,\mathbf{kk'}} \Big[ (\alpha^\dagger_{\mathbf{k}}\beta^\dagger_{\mathbf{k'}} + \beta_{\mathbf{k'}}\alpha_{\mathbf{k}})S_z + \frac{1}{2}(\beta_{\mathbf{k'}}\beta_{\mathbf{k}} - \alpha^\dagger_{\mathbf{k}}\alpha^\dagger_{\mathbf{k'}})S^- + \frac{1}{2}(\beta^\dagger_{\mathbf{k}}\beta^\dagger_{\mathbf{k'}} - \alpha_{\mathbf{k'}}\alpha_{\mathbf{k}})S^+ \Big] \Big\}.
\end{aligned}
\tag{15}
$$

The commutator between $X_{\lambda,\Delta\lambda}$ and $\mathcal{H}_\lambda$ can now be written in the form

$$
[X_{\lambda,\Delta\lambda}, \mathcal{H}_\lambda] = [X_{\lambda,\Delta\lambda}, \mathcal{H}_{0,\lambda}] + [X_{\lambda,\Delta\lambda}, \mathcal{H}_{1,\lambda}].
\tag{16}
$$

Since $\mathcal{H}_{0,\lambda}$ is diagonal in the fermion operators, the first part $[X_{\lambda,\Delta\lambda}, \mathcal{H}_{0,\lambda}]$ is found directly from Eq. (12) by cancellation of the fermion energies $E_{\mathbf{k},\lambda}$ in the denominators. We obtain

$$
\begin{aligned}
[X_{\lambda,\Delta\lambda}, \mathcal{H}_{0,\lambda}] = -\frac{\Delta\lambda}{N} \sum_{\mathbf{k}\neq\mathbf{k'}} &\Big\{ (J_{\mathbf{k},\lambda}+J_{\mathbf{k'},\lambda}) \Big[ C^+_{1,\mathbf{kk'}} \big( \alpha^\dagger_{\mathbf{k}}\alpha_{\mathbf{k'}}S_z - \beta^\dagger_{\mathbf{k}}\beta_{\mathbf{k'}}S_z + \alpha^\dagger_{\mathbf{k}}\beta_{\mathbf{k'}}S^- + \beta^\dagger_{\mathbf{k'}}\alpha_{\mathbf{k}}S^+ \big) \\
&+ C^-_{2,\mathbf{kk'}} \Big( (\alpha^\dagger_{\mathbf{k}}\beta^\dagger_{\mathbf{k'}} + \beta_{\mathbf{k'}}\alpha_{\mathbf{k}})S_z - \frac{1}{2}(-\beta_{\mathbf{k'}}\beta_{\mathbf{k}} + \alpha^\dagger_{\mathbf{k}}\alpha^\dagger_{\mathbf{k'}})S^- + \frac{1}{2}(\beta^\dagger_{\mathbf{k}}\beta^\dagger_{\mathbf{k'}} - \alpha_{\mathbf{k'}}\alpha_{\mathbf{k}})S^+ \Big) \Big] \\
&- (V_{\mathbf{k},\lambda}+V_{\mathbf{k'},\lambda}) \Big[ C^+_{1,\mathbf{kk'}} \big( \alpha^\dagger_{\mathbf{k}}\alpha_{\mathbf{k'}} + \beta^\dagger_{\mathbf{k}}\beta_{\mathbf{k'}} \big) - C^-_{2,\mathbf{kk'}} \big( \alpha^\dagger_{\mathbf{k}}\beta^\dagger_{\mathbf{k'}} + \beta_{\mathbf{k'}}\alpha_{\mathbf{k}} \big) \Big] \Big\}.
\end{aligned}
\tag{17}
$$

The second part of the commutator (16) gives rise to an additional internal summation, and generates new contributions to an effective electron-impurity scattering. We distinguish three types of terms,

$$[X_{\lambda,\Delta\lambda}, \mathcal{H}_{1,\lambda}] = [X_{\lambda,\Delta\lambda}, \mathcal{H}_{1,\lambda}]_1 + [X_{\lambda,\Delta\lambda}, \mathcal{H}_{1,\lambda}]_2 + [X_{\lambda,\Delta\lambda}, \mathcal{H}_{1,\lambda}]_3. \tag{18}$$

The part $[X_{\lambda,\Delta\lambda}, \mathcal{H}_{1,\lambda}]_1$ arises from commutators involving Bogoliobov quasiparticle operators from the first and second lines in Eq. (12). Initially, these terms contain products of spin operators. Due to spin rotation invariance, however, the spin operators combine to form $(\vec{S} \cdot \vec{S})$, and can be replaced by the number $S(S+1)$. The term $[X_{\lambda,\Delta\lambda}, \mathcal{H}_{1,\lambda}]_1$ is therefore merely a bilinear of fermionic operators. We obtain

$$[X_{\lambda,\Delta\lambda}, \mathcal{H}_{1,\lambda}]_1 =$$
$$-\frac{J\lambda\Delta\lambda}{2N} S(S+1) \sum_{\mathbf{k}\mathbf{k}'} \left( A^{(1)}_{\mathbf{k},\lambda} + A^{(1)}_{\mathbf{k}',\lambda} \right) \left[ C^+_{1,\mathbf{k}\mathbf{k}'} \left( \alpha^\dagger_{\mathbf{k}} \alpha_{\mathbf{k}'} + \beta^\dagger_{\mathbf{k}} \beta_{\mathbf{k}'} \right) - C^-_{2,\mathbf{k}\mathbf{k}'} \left( \alpha^\dagger_{\mathbf{k}} \beta^\dagger_{\mathbf{k}'} + \beta_{\mathbf{k}'} \alpha_{\mathbf{k}} \right) \right], \tag{19}$$

where we introduced the short-hand notation

$$A^{(1)}_{\mathbf{k},\lambda} = \frac{1}{N} \sum_{\mathbf{q}(\neq \mathbf{k})} \frac{J_{\mathbf{k},\lambda} + J_{\mathbf{q},\lambda}}{E^2_{\mathbf{q},\lambda} - E^2_{\mathbf{k},\lambda}} \left[ E_{\mathbf{k},\lambda} + \left( C^+_{2,\mathbf{q}\mathbf{q}} - C^-_{1,\mathbf{q}\mathbf{q}} \right) E_{\mathbf{q},\lambda} \right]. \tag{20}$$

Note that according to Eqs. (3) and (6), the coherence factors with respect to equal quantum number $\mathbf{q}$ have the solutions $C^+_{2,\mathbf{q}\mathbf{q}} = \Delta/E_{\mathbf{q}}$ and $C^-_{1,\mathbf{q}\mathbf{q}} = \varepsilon_{\mathbf{q}}/E_{\mathbf{q}}$. It is seen that the operator structure of $[X_{\lambda,\Delta\lambda}, \mathcal{H}_{1,\lambda}]_1$ is different from $\mathcal{H}_{1,\lambda}$ since this commutator part only contains scattering of fermions without a coupling to the spin operator. The natural appearance of such terms is the reason why we have to take into account such decoupled fermion scatterings in the ansatz of the generator in Eq. (12) (third line).

The bilinear terms in the generator give rise to a second type of terms which we denote as $[X_{\lambda,\Delta\lambda}, \mathcal{H}_{1,\lambda}]_2$. It describes the commutator between the third line of Eq. (12) and all of $\mathcal{H}_{1,\lambda}$. We find

$$[X_{\lambda,\Delta\lambda}, \mathcal{H}_{1,\lambda}]_2 = \frac{J\lambda\Delta\lambda}{2N} \sum_{\mathbf{k}\mathbf{k}'} \left( A^{(2)}_{\mathbf{k},\lambda} + \mathcal{A}^{(2)}_{\mathbf{k}',\lambda} \right) \left\{ C^+_{1,\mathbf{k}\mathbf{k}'} \left[ \alpha^\dagger_{\mathbf{k}} \alpha_{\mathbf{k}'} S_z - \beta^\dagger_{\mathbf{k}} \beta_{\mathbf{k}'} S_z + \alpha^\dagger_{\mathbf{k}} \beta_{\mathbf{k}'} S^- + \beta^\dagger_{\mathbf{k}'} \alpha_{\mathbf{k}} S^+ \right] \right.$$
$$\left. + C^-_{2,\mathbf{k}\mathbf{k}'} \left[ \left( \alpha^\dagger_{\mathbf{k}} \beta^\dagger_{\mathbf{k}'} + \beta_{\mathbf{k}'} \alpha_{\mathbf{k}} \right) S_z + \frac{1}{2} \left( \beta_{\mathbf{k}'} \beta_{\mathbf{k}} - \alpha^\dagger_{\mathbf{k}} \alpha^\dagger_{\mathbf{k}'} \right) S^- + \frac{1}{2} \left( \beta^\dagger_{\mathbf{k}} \beta^\dagger_{\mathbf{k}'} - \alpha_{\mathbf{k}'} \alpha_{\mathbf{k}} \right) S^+ \right] \right\}, \tag{21}$$

where we have introduced

$$A^{(2)}_{\mathbf{k},\lambda} = \frac{1}{N} \sum_{\mathbf{q}(\neq \mathbf{k})} \frac{V_{\mathbf{k},\lambda} + V_{\mathbf{q},\lambda}}{E^2_{\mathbf{q},\lambda} - E^2_{\mathbf{k},\lambda}} \left[ E_{\mathbf{k},\lambda} + \left( C^+_{2,\mathbf{q}\mathbf{q}} - C^-_{1,\mathbf{q}\mathbf{q}} \right) E_{\mathbf{q},\lambda} \right]. \tag{22}$$

It is seen that the structure of the original scattering interaction is maintained.

The remaining contributions, summarised in the term $[X_{\lambda,\Delta\lambda}, \mathcal{H}_{1,\lambda}]_3$, contain all terms in which commutators between the spin operators are taken. These new terms would be absent in the case of a classical spin. They describe additional many-body interactions, and are a

crucial new ingredient in our approach. Concretely, we find the following operator structure

$$
\begin{aligned}
[X_{\lambda,\Delta\lambda}, \mathcal{H}_{1,\lambda}]_3 = {} & \frac{J\lambda\Delta\lambda}{2N^2} \sum_{\mathbf{kk}'} \sum_{\mathbf{k}_1 \mathbf{k}_1'} \left\{ \left( \frac{J_{\mathbf{k},\lambda} + J_{\mathbf{k}',\lambda}}{E_{\mathbf{k},\lambda} - E_{\mathbf{k}',\lambda}} + \frac{J_{\mathbf{k}_1,\lambda} + J_{\mathbf{k}_1',\lambda}}{E_{\mathbf{k}_1,\lambda} - E_{\mathbf{k}_1',\lambda}} \right) C^{+}_{1,\mathbf{kk}'} C^{+}_{1,\mathbf{k}_1 \mathbf{k}_1'} \right. \\
& \times \left[ \left( \alpha_{\mathbf{k}}^{\dagger} \alpha_{\mathbf{k}'} - \beta_{\mathbf{k}}^{\dagger} \beta_{\mathbf{k}'} \right) \left( \beta_{\mathbf{k}_1'}^{\dagger} \alpha_{\mathbf{k}_1} S^{+} + \alpha_{\mathbf{k}_1}^{\dagger} \beta_{\mathbf{k}_1'} S^{-} \right) - 2 \alpha_{\mathbf{k}}^{\dagger} \beta_{\mathbf{k}'} \beta_{\mathbf{k}_1'}^{\dagger} \alpha_{\mathbf{k}_1} S_z \right] \\
& + \left( \frac{J_{\mathbf{k},\lambda} + J_{\mathbf{k}',\lambda}}{E_{\mathbf{k},\lambda} + E_{\mathbf{k}',\lambda}} + \frac{J_{\mathbf{k}_1,\lambda} + J_{\mathbf{k}_1',\lambda}}{E_{\mathbf{k}_1,\lambda} + E_{\mathbf{k}_1',\lambda}} \right) C^{-}_{2,\mathbf{kk}'} C^{-}_{2,\mathbf{k}_2 \mathbf{k}_1'} \\
& \times \left[ \left( \alpha_{\mathbf{k}}^{\dagger} \beta_{\mathbf{k}'}^{\dagger} - \beta_{\mathbf{k}'} \alpha_{\mathbf{k}} \right) \left( \beta_{\mathbf{k}_1'} \beta_{\mathbf{k}_1} S^{+} + \alpha_{\mathbf{k}_1}^{\dagger} \alpha_{\mathbf{k}_1'}^{\dagger} S^{-} \right) - \frac{1}{2} \left( \beta_{\mathbf{k}} \beta_{\mathbf{k}'} + \alpha_{\mathbf{k}'}^{\dagger} \alpha_{\mathbf{k}}^{\dagger} \right) \left( \beta_{\mathbf{k}_1'}^{\dagger} \beta_{\mathbf{k}_1}^{\dagger} - \alpha_{\mathbf{k}_1} \alpha_{\mathbf{k}_1'} \right) S_z \right] \\
& + \left[ \left( \frac{J_{\mathbf{k},\lambda} + J_{\mathbf{k}',\lambda}}{E_{\mathbf{k},\lambda} - E_{\mathbf{k}',\lambda}} + \frac{J_{\mathbf{k}_1,\lambda} + J_{\mathbf{k}_1',\lambda}}{E_{\mathbf{k}_1,\lambda} + E_{\mathbf{k}_1',\lambda}} \right) C^{+}_{1,\mathbf{kk}'} C^{-}_{1,\mathbf{k}_1 \mathbf{k}_1'} + \left( \frac{J_{\mathbf{k},\lambda} + J_{\mathbf{k}',\lambda}}{E_{\mathbf{k},\lambda} + E_{\mathbf{k}',\lambda}} + \frac{J_{\mathbf{k}_1,\lambda} + J_{\mathbf{k}_1',\lambda}}{E_{\mathbf{k}_1,\lambda} - E_{\mathbf{k}_1',\lambda}} \right) C^{-}_{1,\mathbf{kk}'} C^{+}_{1,\mathbf{k}_1 \mathbf{k}_1'} \right] \\
& \left. \times \left[ \left( \alpha_{\mathbf{k}}^{\dagger} \alpha_{\mathbf{k}'} - \beta_{\mathbf{k}}^{\dagger} \beta_{\mathbf{k}'} \right) \left( \beta_{\mathbf{k}_1'} \beta_{\mathbf{k}_1} S^{+} + \alpha_{\mathbf{k}_1}^{\dagger} \alpha_{\mathbf{k}_1'}^{\dagger} S^{-} \right) - \frac{1}{2} \left( \alpha_{\mathbf{k}}^{\dagger} \beta_{\mathbf{k}'} + \beta_{\mathbf{k}'}^{\dagger} \alpha_{\mathbf{k}} \right) \left( \beta_{\mathbf{k}_1'}^{\dagger} \beta_{\mathbf{k}_1}^{\dagger} - \alpha_{\mathbf{k}_1} \alpha_{\mathbf{k}_1'} \right) S_z \right] \right\}.
\end{aligned}
\tag{23}
$$

It is at this step that our method requires an approximation to be made. Namely, we apply a factorization approximation [35] that replaces a product of four fermionic operators by a bilinear times an expectation value. Concretely, the factorization is carried out as follows. Each time a transformation yields a term with four Bogoliubov quasiparticle operators of the form $\alpha^{\dagger}\alpha\beta^{\dagger}\beta$, we make the approximation

$$
\alpha^{\dagger}\alpha\beta^{\dagger}\beta \approx \alpha^{\dagger}\alpha \langle \beta^{\dagger}\beta \rangle_{\lambda} + \langle \alpha^{\dagger}\alpha \rangle_{\lambda} \beta^{\dagger}\beta - \langle \alpha^{\dagger}\alpha \rangle_{\lambda} \langle \beta^{\dagger}\beta \rangle_{\lambda},
\tag{24}
$$

where the $\lambda$-dependent expectation value is given by

$$
\langle \mathcal{A} \rangle_{\lambda} = \frac{Tr\left( \mathcal{A} e^{-\beta \mathcal{H}_{\lambda}} \right)}{Tr\left( e^{-\beta \mathcal{H}_{\lambda}} \right)},
\tag{25}
$$

for an arbitrary operator $\mathcal{A}$. The factor $\beta = 1/(k_B T)$ contains the temperature $T$ and the Boltzmann constant $k_B$. A similar replacement is done for terms with four Bogoliubov operators of the same type ($\alpha^{\dagger}$ or $\beta^{\dagger}$) by taking into account normal ordering. Since the factorization is carried out within the transformation step from $\lambda$ to $\lambda - \Delta\lambda$, the expectation value is formed with respect to $\mathcal{H}_{\lambda}$. This factorization restores an operator structure that is quadratic in fermionic operators, and therefore allows us to derive renormalization equations for $J_{\mathbf{k},\lambda}$ and $A_{\mathbf{k},\lambda}$. The price to pay is that these renormalization equations contain expectation values that have to be calculated self-consistently.

The above factorization approximation looks somewhat similar to a mean field approximation. However, it differs from such a treatment since the factorization is applied at each order of the unitary transformation. The resulting series is summed up to infinite order since all orders have a similar structure. Our approach is thus different from a simple mean field approximation in which a single factorization is applied. As discussed in Ref. [33], our factorization approach has already been employed successfully for a treatment of the Kondo effect in a normal-state metal, which illustrates its non-perturbative power. Moreover, this approximation scheme is the standard technique to treat many-body interactions in common flow equation approaches [31,32]. Applying the factorization approximation to $[X_{\lambda,\Delta\lambda}, \mathcal{H}_{1,\lambda}]_3$, we

obtain

$$
\begin{aligned}
[X_{\lambda,\Delta\lambda}, \mathcal{H}_{1,\lambda}]_{3,\text{factorized}} = \\
\frac{J\lambda\Delta\lambda}{2N} \sum_{\mathbf{kk'}} \left( A_{\mathbf{k},\lambda}^{(3)} + A_{\mathbf{k'},\lambda}^{(3)} \right) \Big\{ C_{1,\mathbf{kk'}}^{+} \left[ \alpha_{\mathbf{k}}^{\dagger}\alpha_{\mathbf{k'}} S_z - \beta_{\mathbf{k}}^{\dagger}\beta_{\mathbf{k'}} S_z + \alpha_{\mathbf{k}}^{\dagger}\beta_{\mathbf{k'}} S^{-} + \beta_{\mathbf{k'}}^{\dagger}\alpha_{\mathbf{k}} S^{+} \right] \\
+ C_{2,\mathbf{kk'}}^{-} \Big[ \left( \alpha_{\mathbf{k}}^{\dagger}\beta_{\mathbf{k'}}^{\dagger} + \beta_{\mathbf{k'}}\alpha_{\mathbf{k}} \right) S_z + \frac{1}{2} \left( \beta_{\mathbf{k'}}\beta_{\mathbf{k}} - \alpha_{\mathbf{k}}^{\dagger}\alpha_{\mathbf{k'}}^{\dagger} \right) S^{-} + \frac{1}{2} \left( \beta_{\mathbf{k}}^{\dagger}\beta_{\mathbf{k'}}^{\dagger} - \alpha_{\mathbf{k'}}\alpha_{\mathbf{k}} \right) S^{+} \Big] \Big\},
\end{aligned}
\tag{26}
$$

where we have introduced

$$
\begin{aligned}
A_{\mathbf{k},\lambda}^{(3)} = \frac{1}{N} \sum_{\mathbf{q}(\neq\mathbf{k}),\mathbf{k'}} \frac{J_{\mathbf{k},\lambda} + J_{\mathbf{q},\lambda}}{E_{\mathbf{q},\lambda}^2 - E_{\mathbf{k},\lambda}^2} \Big[ \left( C_{1,\mathbf{qk'}}^{+} + C_{2,\mathbf{qk'}}^{-} \right) E_{\mathbf{q},\lambda} - \left( C_{2,\mathbf{qk'}}^{+} + C_{1,\mathbf{qk'}}^{-} \right) E_{\mathbf{k},\lambda} \Big] \\
\times \langle \alpha_{\mathbf{q}}\alpha_{\mathbf{k'}}^{\dagger} - \beta_{\mathbf{q}}^{\dagger}\beta_{\mathbf{k'}} + \beta_{\mathbf{q}}^{\dagger}\alpha_{\mathbf{k'}}^{\dagger} + \alpha_{\mathbf{k'}}\beta_{\mathbf{q}} \rangle_{\lambda} .
\end{aligned}
\tag{27}
$$

The effective scattering in Eq. (26), which was obtained after the factorization, has the same operator structure as the scattering part proportional to $\lambda$ in Eq. (10). This property allows us to find equations for the coefficients $J_{\mathbf{k},\lambda}$ and $A_{\mathbf{k},\lambda}$ by adding together all the four commutator parts from Eqs. (17), (19), (21), and (26). From Eq. (13), we find

$$
\mathcal{H}_{\lambda-\Delta\lambda} = \mathcal{H}_{\lambda} + [X_{\lambda,\Delta\lambda}, \mathcal{H}_{0,\lambda}] + [X_{\lambda,\Delta\lambda}, \mathcal{H}_{1,\lambda}]_1 + [X_{\lambda,\Delta\lambda}, \mathcal{H}_{1,\lambda}]_2 + [X_{\lambda,\Delta\lambda}, \mathcal{H}_{1,\lambda}]_{3,\text{factorized}} .
\tag{28}
$$

Using Eq. (10) and the calculated commutators, we compare both sides of the Eq. (28) and obtain the following set of renormalization equations,

$$
-\frac{J}{2} = -J_{\mathbf{k},\lambda} + \frac{J\lambda}{2} \left( A_{\mathbf{k},\lambda}^{(2)} + A_{\mathbf{k},\lambda}^{(3)} \right),
\tag{29}
$$

$$
0 = V_{\mathbf{k},\lambda} + \frac{J\lambda}{2} S(S+1) A_{\mathbf{k},\lambda}^{(1)} .
\tag{30}
$$

This set of linear equations determines the coefficients $J_{\mathbf{k},\lambda}$ and $V_{\mathbf{k},\lambda}$ as a function of $\lambda$. In addition, we find the renormalization equation for $E_{\mathbf{k},\lambda}$,

$$
E_{\mathbf{k},\lambda-\Delta\lambda} = E_{\mathbf{k},\lambda} - \frac{J\lambda\Delta\lambda}{N^2} S(S+1) \sum_{\mathbf{q}} \frac{J_{\mathbf{k},\lambda} + J_{\mathbf{q},\lambda}}{E_{\mathbf{q},\lambda}^2 - E_{\mathbf{k},\lambda}^2} \Big[ E_{\mathbf{k},\lambda} + \left( C_{2,\mathbf{qq}}^{+} - C_{1,\mathbf{qq}}^{-} \right) E_{\mathbf{q},\lambda} \Big] .
\tag{31}
$$

A detailed account of how we solve these equations in practice is given further below. Before turning to implementations, however, let us dwell on the above equations to show how bound states emerge in our approach.

## 2.2 Bound states

From the renormalization equation (31), one finds that in the thermodynamic limit a significant change from $E_{\mathbf{k},\lambda}$ to $E_{\mathbf{k},\lambda-\Delta\lambda}$ only occurs if the coefficient $J_{\mathbf{k},\lambda}$ is singular at a particular $\mathbf{k}$, i. e. if $J_{\mathbf{k},\lambda} \propto N$. The reason is the factor $1/N^2$ which suppresses any renormalization in the thermodynamic limit if $J_{\mathbf{k},\lambda}$ is non-singular. Let us in the following explicitly consider the renormalization equations at such a singular point $\mathbf{k}_0$ and therefore set $\mathbf{k} = \mathbf{k}_0$. Specializing to a system with only a single bound state, we expect that no other energy is renormalized significantly, and therefore set $E_{\mathbf{q},\lambda} = E_{\mathbf{q}}$ in the internal summations in $A_{\mathbf{k}_0,\lambda}^{(1-3)}$. This assumption has been checked to be valid in our explicit implementations. Eqs. (30) and (20) imply that also $A_{\mathbf{k}_0,\lambda}^{(1)}$ and $V_{\mathbf{k}_0,\lambda}$ become singular at $\mathbf{k}_0$. Moreover, we can neglect $J_{\mathbf{q},\lambda}$ and $V_{\mathbf{q},\lambda}$ with $\mathbf{q} \neq \mathbf{k}_0$

compared to $J_{\mathbf{k}_0,\lambda}$ and $V_{\mathbf{k}_0,\lambda}$. Taking into account all these simplifications, Eqs. (29) and (30) at $\mathbf{k}_0$ become

$$
\begin{aligned}
-\frac{J}{2} = {} & -J_{\mathbf{k}_0,\lambda} + \frac{J\lambda}{2}\frac{V_{\mathbf{k}_0,\lambda}}{N}\sum_{\mathbf{q}(\neq \mathbf{k}_0)}\frac{E_{\mathbf{k}_0,\lambda}+\Delta-\varepsilon_{\mathbf{q}}}{E_{\mathbf{q}}^2-E_{\mathbf{k}_0,\lambda}^2} \\
& + \frac{J\lambda}{2}\frac{J_{\mathbf{k}_0,\lambda}}{N}\sum_{\mathbf{q}(\neq \mathbf{k}),\mathbf{k}'}\frac{\left(C_{1,\mathbf{q}\mathbf{k}'}^{+}+C_{2,\mathbf{q}\mathbf{k}'}^{-}\right)E_{\mathbf{q}}-\left(C_{2,\mathbf{q}\mathbf{k}'}^{+}+C_{1,\mathbf{q}\mathbf{k}'}^{-}\right)E_{\mathbf{k}_0,\lambda}}{E_{\mathbf{q}}^2-E_{\mathbf{k}_0,\lambda}^2} \\
& \times \langle \alpha_{\mathbf{q}}\alpha_{\mathbf{k}'}^{\dagger} - \beta_{\mathbf{q}}^{\dagger}\beta_{\mathbf{k}'} + \beta_{\mathbf{q}}^{\dagger}\alpha_{\mathbf{k}'}^{\dagger} + \alpha_{\mathbf{k}'}\beta_{\mathbf{q}}\rangle_{\lambda},
\end{aligned}
\tag{32}
$$

$$
V_{\mathbf{k}_0,\lambda} = -\frac{J\lambda}{2}S(S+1)\frac{J_{\mathbf{k}_0,\lambda}}{N}\sum_{\mathbf{q}(\neq \mathbf{k}_0)}\frac{E_{\mathbf{k}_0,\lambda}+\Delta-\varepsilon_{\mathbf{q}}}{E_{\mathbf{q}}^2-E_{\mathbf{k}_0,\lambda}^2}.
\tag{33}
$$

Let us now consider at first the renormalization in the very first renormalization step from $\lambda = 1$ to $\lambda = 1 - \Delta\lambda$ ($\Delta\lambda \ll 1$), where we can set the $\lambda$ factor equal to 1. Combining the two equations by elimination of $V_{\mathbf{k}_0,\lambda}$ and solving the resulting equation for $J_{\mathbf{k}_0,\lambda}$, we obtain

$$
\begin{aligned}
J_{\mathbf{k}_0,\lambda=1} = {} & \frac{J}{2}\Bigg[ 1 - S(S+1)\left(\frac{J}{2}\frac{1}{N}\sum_{\mathbf{q}(\neq \mathbf{k}_0)}\frac{E_{\mathbf{k}_0,\lambda=1}+\Delta-\varepsilon_{\mathbf{q}}}{E_{\mathbf{q}}^2-E_{\mathbf{k}_0,\lambda=1}^2}\right)^2 \\
& + \frac{J}{2}\frac{1}{N}\sum_{\mathbf{q}(\neq \mathbf{k}_0),\mathbf{k}'}\frac{\left(C_{1,\mathbf{q}\mathbf{k}'}^{+}+C_{2,\mathbf{q}\mathbf{k}'}^{-}\right)E_{\mathbf{q}}-\left(C_{2,\mathbf{q}\mathbf{k}'}^{+}+C_{1,\mathbf{q}\mathbf{k}'}^{-}\right)E_{\mathbf{k}_0,\lambda=1}}{E_{\mathbf{q}}^2-E_{\mathbf{k}_0,\lambda=1}^2} \\
& \times \langle \alpha_{\mathbf{q}}\alpha_{\mathbf{k}'}^{\dagger} - \beta_{\mathbf{q}}^{\dagger}\beta_{\mathbf{k}'} + \beta_{\mathbf{q}}^{\dagger}\alpha_{\mathbf{k}'}^{\dagger} + \alpha_{\mathbf{k}'}\beta_{\mathbf{q}}\rangle\Bigg]^{-1}.
\end{aligned}
\tag{34}
$$

Note that the expectation value now also refers to $\lambda = 1$, i. e. according to Eq. (25) it is taken with the full Hamiltonian $\mathcal{H}$. We therefore omitted the index $\lambda$ in the expectation value. According to Eq. (34), a singularity of $J_{\mathbf{k}_0,\lambda=1}$, which in turn implies a finite renormalization of $E_{\mathbf{k}_0,\lambda=1}$, is found if the equation

$$
\begin{aligned}
1 = {} & S(S+1)\left(\frac{J}{2}\frac{1}{N}\sum_{\mathbf{q}(\neq \mathbf{k}_0)}\frac{E_{\mathbf{k}_0,\lambda=1}+\Delta-\varepsilon_{\mathbf{q}}}{E_{\mathbf{q}}^2-E_{\mathbf{k}_0,\lambda=1}^2}\right)^2 \\
& + \frac{J}{2}\frac{1}{N}\sum_{\mathbf{q}(\neq \mathbf{k}_0),\mathbf{k}'}\frac{\left(C_{1,\mathbf{q}\mathbf{k}'}^{+}+C_{2,\mathbf{q}\mathbf{k}'}^{-}\right)E_{\mathbf{q}}-\left(C_{2,\mathbf{q}\mathbf{k}'}^{+}+C_{1,\mathbf{q}\mathbf{k}'}^{-}\right)E_{\mathbf{k}_0,\lambda=1}}{E_{\mathbf{q}}^2-E_{\mathbf{k}_0,\lambda=1}^2}\langle \alpha_{\mathbf{q}}\alpha_{\mathbf{k}'}^{\dagger} - \beta_{\mathbf{q}}^{\dagger}\beta_{\mathbf{k}'} + \beta_{\mathbf{q}}^{\dagger}\alpha_{\mathbf{k}'}^{\dagger} + \alpha_{\mathbf{k}'}\beta_{\mathbf{q}}\rangle
\end{aligned}
\tag{35}
$$

is fulfilled. The solution $E_{\mathbf{k}_0,\lambda=1}$ of this equation is the renormalized energy value due to the singularity. Note that the summation over the expectation values in the second line in general prevents an analytical solution. Therefore, this equation must be solved self-consistently. Only in the special case of a classical spin, where the second line is absent, is an analytical solution possible and yields the well-known YSR states (see Appendix A). In general, one finds two solutions of Eq. (35), where the first solution is a part of the quasiparticle band, i.e. $E_{\mathbf{k}_0} \geq \Delta$, and the second solution lies inside the gap, $E_{\mathbf{k}_0,\lambda} < \Delta$. It follows that the renormalization starts at this particular solution $E_{\mathbf{k}_0}$ and ends up with the second solution $E_{\mathbf{k}_0,\lambda}$ at some finite value of $\lambda$. All other states at $\mathbf{q} \neq \mathbf{k}_0$ are unaffected and the corresponding energies remain unchanged, $E_{\mathbf{q},\lambda} = E_{\mathbf{q}}$.

In summary, at the starting point $\lambda = 1$, a particular state with quantum number $\mathbf{k}_0$ from the quasiparticle band experiences a finite renormalization as the corresponding transformation coefficients $J_{\mathbf{k}_0,\lambda}, V_{\mathbf{k}_0,\lambda}$ become singular, i. e. they take values proportional to $N$. Finally,

we emphasise that the formation of a bound state as described here is a general property of an impurity which is embedded in a bath of fermions. This is shown in Appendix B for a toy model of a non-magnetic impurity which is coupled to spinless fermions.

## 2.3 Expectation values

In addition to the energy parameters considered so far also a renormalization equation for the $\lambda$-dependent expectation value in Eq. (27) has to be found. In this expectation value, the quantum numbers $\mathbf{k}'$ and $\mathbf{q}$ may also be equal (the expectation value then describes an occupation number of the Bogoliubov quasiparticles). Let us begin by considering the expectation value $\langle \alpha_{\mathbf{k}}^\dagger \alpha_{\mathbf{q}} \rangle_\lambda$ for the $\alpha$-type of the two quasiparticles (the corresponding expectation value of the $\beta$ type can be found analogously). To derive the $\lambda$ dependence for this expectation value, we use that the trace of any operator is invariant under a unitary transformation. According to Eq. (25), we can thus write

$$\langle \alpha_{\mathbf{k}}^\dagger \alpha_{\mathbf{q}} \rangle_\lambda = \langle e^{X_{\lambda,\Delta\lambda}} \alpha_{\mathbf{k}}^\dagger e^{-X_{\lambda,\Delta\lambda}} e^{X_{\lambda,\Delta\lambda}} \alpha_{\mathbf{q}} e^{-X_{\lambda,\Delta\lambda}} \rangle_{\lambda-\Delta\lambda}, \tag{36}$$

where Eq. (11) was used. The transformed single-particle operator up to linear order in the small parameter $\Delta\lambda \ll 1$ is found from the expression (12) of $X_{\lambda,\Delta\lambda}$,

$$
\begin{aligned}
e^{X_{\lambda,\Delta\lambda}} \alpha_{\mathbf{k}}^\dagger e^{-X_{\lambda,\Delta\lambda}} \approx{} & \alpha_{\mathbf{k}}^\dagger + \frac{\Delta\lambda}{N} \sum_{\mathbf{k}'} \Bigg\{ \left( J_{\mathbf{k},\lambda} + J_{\mathbf{k}',\lambda} \right) \left[ \frac{C_{1,\mathbf{k}'\mathbf{k}}^+}{E_{\mathbf{k}',\lambda} - E_{\mathbf{k},\lambda}} \left( \alpha_{\mathbf{k}'}^\dagger S_z - \beta_{\mathbf{k}'}^\dagger \alpha_{\mathbf{k}} S^+ \right) \right. \\
& \left. - \frac{C_{2,\mathbf{k}\mathbf{k}'}^-}{E_{\mathbf{k},\lambda} + E_{\mathbf{k}',\lambda}} \left( \beta_{\mathbf{k}'} S_z - \alpha_{\mathbf{k}'} S^+ \right) \right] + \left( V_{\mathbf{k},\lambda} + V_{\mathbf{k}',\lambda} \right) \left[ \frac{C_{1,\mathbf{k}\mathbf{k}'}^+}{E_{\mathbf{k},\lambda} - E_{\mathbf{k}',\lambda}} \alpha_{\mathbf{k}'}^\dagger - \frac{C_{2,\mathbf{k}\mathbf{k}'}^-}{E_{\mathbf{k},\lambda} + E_{\mathbf{k}',\lambda}} \beta_{\mathbf{k}'} \right] \Bigg\}.
\end{aligned}
\tag{37}
$$

Plugging this expression into Eq. (36) we find the following relation between the expectation values at $\lambda$ and the ones at the reduced parameter $\lambda - \Delta\lambda$,

$$
\begin{aligned}
\langle \alpha_{\mathbf{k}}^\dagger \alpha_{\mathbf{q}} \rangle_\lambda \approx{} & \langle \alpha_{\mathbf{k}}^\dagger \alpha_{\mathbf{q}} \rangle_{\lambda-\Delta\lambda} \\
& + \frac{\Delta\lambda}{N} \sum_{\mathbf{k}'} \left( V_{\mathbf{q},\lambda} + V_{\mathbf{k}',\lambda} \right) \left( \frac{C_{1,\mathbf{q}\mathbf{k}'}^+}{E_{\mathbf{q},\lambda} - E_{\mathbf{k}',\lambda}} \langle \alpha_{\mathbf{k}}^\dagger \alpha_{\mathbf{k}'} \rangle_{\lambda-\Delta\lambda} - \frac{C_{2,\mathbf{q}\mathbf{k}'}^-}{E_{\mathbf{q},\lambda} + E_{\mathbf{k}',\lambda}} \langle \alpha_{\mathbf{k}}^\dagger \beta_{\mathbf{k}'}^\dagger \rangle_{\lambda-\Delta\lambda} \right) \\
& + \frac{\Delta\lambda}{N} \sum_{\mathbf{k}'} \left( V_{\mathbf{k},\lambda} + V_{\mathbf{k}',\lambda} \right) \left( \frac{C_{1,\mathbf{k}\mathbf{k}'}^+}{E_{\mathbf{k},\lambda} - E_{\mathbf{k}',\lambda}} \langle \alpha_{\mathbf{k}'}^\dagger \alpha_{\mathbf{q}} \rangle_{\lambda-\Delta\lambda} - \frac{C_{2,\mathbf{k}\mathbf{k}'}^-}{E_{\mathbf{k},\lambda} + E_{\mathbf{k}',\lambda}} \langle \beta_{\mathbf{k}'} \alpha_{\mathbf{q}} \rangle_{\lambda-\Delta\lambda} \right).
\end{aligned}
\tag{38}
$$

Similar equations describe the $\lambda$-dependence of the anomalous expectation values $\langle \alpha_{\mathbf{k}}^\dagger \beta_{\mathbf{q}}^\dagger \rangle_\lambda$, $\langle \beta_{\mathbf{q}} \alpha_{\mathbf{k}} \rangle_\lambda$. The above set of linear equations provides a system of renormalization equations for the $\lambda$-dependent expectation values. The $\lambda$-dependence becomes particularly significant if one of the transformation coefficients $J_{\mathbf{k},\lambda}$ and $V_{\mathbf{k},\lambda}$ becomes singular ($\propto N$), which is the case for the bound state $\mathbf{k}_0$ (see previous section). The numerical evaluation of the renormalization equations is carried out in parallel to the equations (29)-(31) starting from $\lambda = 1$ down to $\lambda = 0$. The starting value $\langle \alpha_{\mathbf{k}}^\dagger \alpha_{\mathbf{q}} \rangle_{\lambda=1} = \langle \alpha_{\mathbf{k}}^\dagger \alpha_{\mathbf{q}} \rangle$ is the scattering probability with respect to the original Hamiltonian. This quantitiy has to be calculated self-consistently as explained in Sec. 3.

In addition to appearing in the renormalization equations, expectation values are also central for physical observables. We therefore conclude this technical section by detailing the general procedure of calculating expectation values in our approach. We are interested in the expectation value of some operator $\mathcal{A}$ formed with respect to the original Hamiltonian. Formally, this corresponds to Eq. (25) at $\lambda = 1$,

$$\langle \mathcal{A} \rangle = \langle \mathcal{A} \rangle_{\lambda=1} = \frac{Tr \left( \mathcal{A} e^{-\beta\mathcal{H}} \right)}{Tr \left( e^{-\beta\mathcal{H}} \right)}. \tag{39}$$

To evaluate Eq. (39), we use the invariance of the trace with respect to unitary transformations and rewrite Eq. (39) introducing the stepwise unitary transformations down to $\lambda = 0$, where the effective Hamiltonian $\mathcal{H}_{\lambda=0} = \tilde{\mathcal{H}}$ is diagonal, see Eq. (9). We find

$$\langle \mathcal{A} \rangle = \frac{Tr\left( \tilde{\mathcal{A}} e^{-\beta\tilde{\mathcal{H}}} \right)}{Tr\left( e^{-\beta\tilde{\mathcal{H}}} \right)} . \tag{40}$$

The practical calculation of observables therefore requires the transformation of the associated operator $\mathcal{A}$ using the same procedure as carried out to find $\tilde{\mathcal{H}}$. To simplify the discussion, let us discuss this method using the example of the occupation number $\langle \alpha_{\mathbf{k}}^{\dagger} \alpha_{\mathbf{k}} \rangle$, which is the quantity entering all renormalization equations. Eq. (40) then reads

$$\langle \alpha_{\mathbf{k}}^{\dagger} \alpha_{\mathbf{k}} \rangle = \frac{Tr\left( \tilde{\alpha}_{\mathbf{k}}^{\dagger} \tilde{\alpha}_{\mathbf{k}} e^{-\beta\tilde{\mathcal{H}}} \right)}{Tr\left( e^{-\beta\tilde{\mathcal{H}}} \right)} . \tag{41}$$

Similar expressions can be derived for any other expectation value. The main task is therefore to find the transformed operators $\tilde{\alpha}_{\mathbf{k}}^{\dagger}$ by applying the same series of unitary transformation that was used to transform the Hamiltonian. As for the latter, this results in $\lambda$-dependent renormalizations of $\tilde{\alpha}_{\mathbf{k}}^{\dagger}$. Eq. (37) suggests the ansatz

$$\alpha_{\mathbf{k}\lambda}^{\dagger} = x_{\mathbf{k}\lambda} \alpha_{\mathbf{k}}^{\dagger} + \frac{y_{\mathbf{k}\lambda}}{N} \sum_{\mathbf{k}'(\neq\mathbf{k})} \left\{ \left( J_{\mathbf{k},\lambda} + J_{\mathbf{k}',\lambda} \right) \left[ \frac{C_{1,\mathbf{k}'\mathbf{k}}^{+}}{E_{\mathbf{k}',\lambda} - E_{\mathbf{k},\lambda}} \left( \alpha_{\mathbf{k}'}^{\dagger} S_z - \beta_{\mathbf{k}'}^{\dagger} S^{+} \right) \right. \right.$$
$$\left. - \frac{C_{2,\mathbf{k}\mathbf{k}'}^{-}}{E_{\mathbf{k},\lambda} + E_{\mathbf{k}',\lambda}} \left( \beta_{\mathbf{k}'} S_z - \alpha_{\mathbf{k}'} S^{+} \right) \right] - \left( V_{\mathbf{k},\lambda} + V_{\mathbf{k}',\lambda} \right) \left[ \frac{C_{1,\mathbf{k}\mathbf{k}'}^{+}}{E_{\mathbf{k}',\lambda} - E_{\mathbf{k},\lambda}} \alpha_{\mathbf{k}'}^{\dagger} + \frac{C_{2,\mathbf{k}\mathbf{k}'}^{-}}{E_{\mathbf{k},\lambda} + E_{\mathbf{k}',\lambda}} \beta_{\mathbf{k}'} \right] \right\} , \tag{42}$$

where the initial conditions at $\lambda = 1$ are $x_{\mathbf{k}\lambda=1} = 1$ and $y_{\mathbf{k}\lambda=1} = 0$. The renormalization equations for the $\lambda$-dependent coefficients $x_{\mathbf{k}\lambda}$ and $y_{\mathbf{k}\lambda}$ in Eq. (42) can be found by evaluating a similar equation as Eq. (11) but for the renormalized single particle operator,

$$\alpha_{\mathbf{k}\lambda-\Delta\lambda}^{\dagger} \approx e^{X_{\lambda,\Delta\lambda}} \alpha_{\mathbf{k}\lambda}^{\dagger} e^{-X_{\lambda,\Delta\lambda}}, \tag{43}$$

where Eq. (12) must be used for $X_{\lambda,\Delta\lambda}$. Evaluating again the right hand side up to the linear order in $\Delta\lambda \ll 1$ and using the above ansatz for $\alpha_{\mathbf{k}\lambda}^{\dagger}$, we find

$$x_{\mathbf{k},\lambda-\Delta\lambda} = x_{\mathbf{k},\lambda} - y_{\mathbf{k}\lambda} \frac{\Delta\lambda}{N^2} \sum_{\mathbf{q}(\neq\mathbf{k})} \left[ S(S+1) \left( J_{\mathbf{k},\lambda} + J_{\mathbf{q},\lambda} \right)^2 + \left( V_{\mathbf{k},\lambda} + V_{\mathbf{q},\lambda} \right)^2 \right]$$
$$\times \left[ \left( \frac{C_{1,\mathbf{kq}}^{+}}{E_{\mathbf{k},\lambda} - E_{\mathbf{q},\lambda}} \right)^2 + \left( \frac{C_{2,\mathbf{kq}}^{-}}{E_{\mathbf{k},\lambda} + E_{\mathbf{q},\lambda}} \right)^2 \right] . \tag{44}$$

The corresponding equation for $y_{\mathbf{k},\lambda}$ is found from a sum rule which must be fulfilled at each $\lambda$,

$$1 = x_{\mathbf{k}\lambda}^2 + y_{\mathbf{k}\lambda}^2 \frac{1}{N^2} \sum_{\mathbf{q}(\neq\mathbf{k})} \left[ S(S+1) \left( J_{\mathbf{k},\lambda} + J_{\mathbf{q},\lambda} \right)^2 + \left( V_{\mathbf{k},\lambda} + V_{\mathbf{q},\lambda} \right)^2 \right]$$
$$\times \left[ \left( \frac{C_{1,\mathbf{kq}}^{+}}{E_{\mathbf{k},\lambda} - E_{\mathbf{q},\lambda}} \right)^2 + \left( \frac{C_{2,\mathbf{kq}}^{-}}{E_{\mathbf{k},\lambda} + E_{\mathbf{q},\lambda}} \right)^2 \right] . \tag{45}$$

This sum rule results from the requirement that $[\alpha_{\mathbf{k}\lambda}^{\dagger}, \alpha_{\mathbf{k}\lambda}]_{+} = 1$ holds. We find

$$
y_{\mathbf{k},\lambda-\Delta\lambda}^2 = \left(1 - x_{\mathbf{k}\lambda-\Delta\lambda}^2\right) / \frac{1}{N^2} \sum_{\mathbf{q}(\neq\mathbf{k})} \left[ S(S+1)\left(J_{\mathbf{k},\lambda-\Delta\lambda} + J_{\mathbf{q},\lambda-\Delta\lambda}\right)^2 + \left(V_{\mathbf{k},\lambda-\Delta\lambda} + V_{\mathbf{q},\lambda-\Delta\lambda}\right)^2 \right]
$$
$$
\times \left[ \left( \frac{C_{1,\mathbf{kq}}^{+}}{E_{\mathbf{k},\lambda-\Delta\lambda} - E_{\mathbf{q},\lambda-\Delta\lambda}} \right)^2 + \left( \frac{C_{2,\mathbf{kq}}^{-}}{E_{\mathbf{k},\lambda-\Delta\lambda} + E_{\mathbf{q},\lambda-\Delta\lambda}} \right)^2 \right].
$$
(46)

To calculate expectation values, the set of renormalization equations (44) and (46) has to be solved simultaneous with the corresponding equations for the Hamiltonian up to $\lambda = 0$. This yields the fully renormalized single-particle operator at $\lambda = 0$ as obtained from Eq. (42),

$$
\tilde{\alpha}_{\mathbf{k}}^{\dagger} = \tilde{x}_{\mathbf{k}} \alpha_{\mathbf{k}}^{\dagger} + \frac{\tilde{y}_{\mathbf{k}}}{N} \sum_{\mathbf{k}'(\neq\mathbf{k})} \left\{ (\tilde{J}_{\mathbf{k}} + \tilde{J}_{\mathbf{k}'}) \left[ \frac{C_{1,\mathbf{k}'\mathbf{k}}^{+}}{\tilde{E}_{\mathbf{k}'} - \tilde{E}_{\mathbf{k}}} \left( \alpha_{\mathbf{k}'}^{\dagger} S_z - \beta_{\mathbf{k}'}^{\dagger} S^{+} \right) - \frac{C_{2,\mathbf{kk}'}^{-}}{\tilde{E}_{\mathbf{k}} + \tilde{E}_{\mathbf{k}'}} \left( \beta_{\mathbf{k}'} S_z - \alpha_{\mathbf{k}'} S^{+} \right) \right] \right.
$$
$$
\left. - (\tilde{V}_{\mathbf{k}} + \tilde{V}_{\mathbf{k}'}) \left[ \frac{C_{1,\mathbf{kk}'}^{+}}{\tilde{E}_{\mathbf{k}'} - \tilde{E}_{\mathbf{k}}} \alpha_{\mathbf{k}'}^{\dagger} + \frac{C_{2,\mathbf{kk}'}^{-}}{\tilde{E}_{\mathbf{k}} + \tilde{E}_{\mathbf{k}'}} \beta_{\mathbf{k}'} \right] \right\}.
$$
(47)

Note that a possible bound state $\mathbf{k}_0$ may be also included in the summation. If we consider a state with $\mathbf{k} \neq \mathbf{k}_0$, we find

$$
\tilde{\alpha}_{\mathbf{k}}^{\dagger} = \tilde{x}_{\mathbf{k}} \alpha_{\mathbf{k}}^{\dagger} + \frac{\tilde{y}_{\mathbf{k}}}{N} \sum_{\mathbf{k}'(\neq\mathbf{k},\mathbf{k}_0)} \left\{ (\tilde{J}_{\mathbf{k}} + \tilde{J}_{\mathbf{k}'}) \left[ \frac{C_{1,\mathbf{k}'\mathbf{k}}^{+}}{E_{\mathbf{k}'} - E_{\mathbf{k}}} \left( \alpha_{\mathbf{k}'}^{\dagger} S_z - \beta_{\mathbf{k}'}^{\dagger} S^{+} \right) - \frac{C_{2,\mathbf{kk}'}^{-}}{E_{\mathbf{k}} + E_{\mathbf{k}'}} \left( \beta_{\mathbf{k}'} S_z - \alpha_{\mathbf{k}'} S^{+} \right) \right] \right.
$$
$$
\left. - (\tilde{V}_{\mathbf{k}} + \tilde{V}_{\mathbf{k}'}) \left[ \frac{C_{1,\mathbf{kk}'}^{+}}{E_{\mathbf{k}'} - E_{\mathbf{k}}} \alpha_{\mathbf{k}'}^{\dagger} + \frac{C_{2,\mathbf{kk}'}^{-}}{E_{\mathbf{k}} + E_{\mathbf{k}'}} \beta_{\mathbf{k}'} \right] \right\}
$$
$$
+ \frac{C_{1,\mathbf{k}_0\mathbf{k}}^{+}}{\tilde{E}_{\mathbf{k}_0} - E_{\mathbf{k}}} \left( \tilde{J} \alpha_{\mathbf{k}_0}^{\dagger} S_z - \tilde{J} \beta_{\mathbf{k}_0}^{\dagger} S^{+} - \tilde{V} \alpha_{\mathbf{k}_0}^{\dagger} \right) - \frac{C_{2,\mathbf{kk}_0}^{-}}{E_{\mathbf{k}} + \tilde{E}_{\mathbf{k}_0}} \left( \tilde{J} \beta_{\mathbf{k}_0} S_z - \tilde{J} \alpha_{\mathbf{k}_0} S^{+} + \tilde{V} \beta_{\mathbf{k}_0} \right),
$$
(48)

where the parameters $\tilde{J}$ and $\tilde{V}$ are the renormalized coupling parameters of the bound state. They fulfil the relations $\tilde{J}_{\mathbf{k}_0} = N\tilde{J}$ and $\tilde{V}_{\mathbf{k}_0} = N\tilde{V}$. It is seen that the renormalized quasiparticle operator consists of three contributions. The first part, proportional to $\tilde{x}_{\mathbf{k}}$, describes a coherent excitation reminiscent of the original superconducting quasiparticles. As we show in Eq. (54) below, this part of the renormalized quasiparticle operator also lives at the energy $E_{\mathbf{k}}$. This implies that the quantum number $\mathbf{k}$ can then still be interpreted as a momentum (after all, we consider a system with a single impurity, such that the breaking of translational invariance by the localized impurity does not dramatically impact many of the states in our bulk system). The second part, proportional to $\tilde{y}_{\mathbf{k}}$, results from the scattering of an incoming quasiparticle at $\mathbf{k}$ into an outgoing state $\mathbf{k}' \neq \mathbf{k}, \mathbf{k}_0$. This part describes a continuum of excitations which is governed by the renormalized exchange coupling $\tilde{J}_{\mathbf{k}}$. Its operator strucutre, e.g. $\alpha_{\mathbf{k}'}^{\dagger} S_z - \beta_{\mathbf{k}'}^{\dagger} S^{+}$, is well-known in the context of Kondo physics [39], where operators of this form describe composite fermion excitations. As we further explain below, these terms indeed correspond to a Kondo resonance. The remaining terms, finally, are associated with $\mathbf{k}_0$, and therefore describe the formation of a YSR bound state appearing at the (renormalized) energy $\tilde{E}_{\mathbf{k}_0}$ inside the superconducting gap.

To illustrate this general scheme, let us consider two specific examples of expectation values which are needed for our discussions. First, we consider the static occupation number of the

superconducting quasiparticles. Using Eqs. (41) and (48), the occupation number reads

$$
\begin{aligned}
\langle a_{\mathbf{k}}^{\dagger} a_{\mathbf{k}} \rangle = {}& \check{x}_{\mathbf{k}}^{2} f(\tilde{E}_{\mathbf{k}}) + \check{y}_{\mathbf{k}}^{2} \frac{1}{N^{2}} \sum_{\mathbf{q}(\neq \mathbf{k}, \mathbf{k}_0)} \left[ S(S+1)\left(\tilde{J}_{\mathbf{k}} + \tilde{J}_{\mathbf{q}}\right)^{2} + \left(\tilde{V}_{\mathbf{k}} + \tilde{V}_{\mathbf{q}}\right)^{2} \right] \\
& \times \left[ \left(\frac{C_{1,\mathbf{kq}}^{+}}{E_{\mathbf{k}} - E_{\mathbf{q}}}\right)^{2} f(E_{\mathbf{q}}) + \left(\frac{C_{2,\mathbf{kq}}^{-}}{E_{\mathbf{k}} + E_{\mathbf{q}}}\right)^{2} \left(1 - f(E_{\mathbf{q}})\right) \right] \\
& + \left[ S(S+1)\tilde{J}^{2} + \tilde{V}^{2} \right] \left[ \left(\frac{C_{1,\mathbf{kk}_0}^{+}}{E_{\mathbf{k}} - \tilde{E}_{\mathbf{k}_0}}\right)^{2} f(\tilde{E}_{\mathbf{k}_0}) + \left(\frac{C_{2,\mathbf{kk}_0}^{-}}{E_{\mathbf{k}} + \tilde{E}_{\mathbf{k}_0}}\right)^{2} \left(1 - f(\tilde{E}_{\mathbf{k}_0})\right) \right],
\end{aligned}
\tag{49}
$$

where $f(E) = 1/(1 + e^{\beta E})$ is the Fermi function with respect to an energy $E$.

As a second example, we turn to the single-particle spectral function of the original electrons as a function of $\mathbf{k}$. In the absence of a coupling to the quantum spin, $J = 0$, this quantum number corresponds to the electron's momentum. The presence of the impurity spin breaks translational invariance, such that $\mathbf{k}$ strictly speaking cannot be interpreted as a momentum anymore. However, as argued above and illustrated by Eqs. (48) and (54), most of the states in the considered bulk system are not affected by the presence of a single impurity. We can therefore in practice understand $\mathbf{k}$ as a quantum number that for most states corresponds to momentum. The single-particle spectral function as a function of $\mathbf{k}$ in that sense still allows to visualize the excitation structure in momentum space. It is defined via the imaginary part of the single-particle Green's function,

$$
\Im G(\mathbf{k}, \mathbf{k}, \omega) = \frac{1}{2\pi} \int_{-\infty}^{\infty} \langle [c_{\mathbf{k}\sigma}^{\dagger}(-t), c_{\mathbf{k}\sigma}]_{+} \rangle e^{i\omega t} dt.
\tag{50}
$$

The time-dependence and the expectation value is calculated using the original Hamiltonian $\mathcal{H}$, i. e. the expectation value is defined as in Eq. (39). Using the Mori-Zwanzig formalism [37, 38], the anticommutator correlation function can be at first rewritten to the form

$$
\Im G(\mathbf{k}, \mathbf{k}, \omega) = \langle [c_{\mathbf{k}\sigma}, \delta(\mathbf{L} - \omega) c_{\mathbf{k}\sigma}^{\dagger}]_{+} \rangle,
\tag{51}
$$

where $\mathbf{L}\mathcal{A} = [\mathcal{H}, \mathcal{A}]$ is the Liouville operator. For occupied states $\mathbf{k}$, this correlation function can be probed by angle resolved photo emission (ARPES). Our method to calculate this quantity is by using the invariance of the trace under the unitary transformation as explained above in the context of Eqs. (40) and (41). This yields

$$
\Im G(\mathbf{k}, \mathbf{k}, \omega) = \langle [\tilde{c}_{\mathbf{k}\sigma}, \delta(\tilde{\mathbf{L}} - \omega) \tilde{c}_{\mathbf{k}\sigma}^{\dagger}]_{+} \rangle.
\tag{52}
$$

We evaluate the Liouville operator and the expectation value by replacing the electron operators with the Bogoliubov quasiparticles using Eq. (2) and obtain

$$
\Im G(\mathbf{k}, \mathbf{k}, \omega) = \langle [u_{\mathbf{k}} \tilde{\alpha}_{\mathbf{k}} + v_{\mathbf{k}} \tilde{\beta}_{\mathbf{k}}^{\dagger}, \delta(\tilde{\mathbf{L}} - \omega)(u_{\mathbf{k}} \tilde{\alpha}_{\mathbf{k}}^{\dagger} + v_{\mathbf{k}} \tilde{\beta}_{\mathbf{k}})]_{+} \rangle,
\tag{53}
$$

where for the sake of concreteness, we have here considered the spin-up case. Finally, it remains to replace the fully renormalized single-particle operators $\tilde{\alpha}_{\mathbf{k}}$ with Eq. (48) and the corresponding equation for $\tilde{\beta}_{\mathbf{k}}$ (not explicitly given). Since the effective Hamiltonian $\tilde{\mathcal{H}}$ which is used to calculate the expectation value is diagonal with respect to the $\alpha$ and $\beta$ operators,

one can simply replace the Liouville operator with the corresponding eigenenergy. We obtain

$$
\Im G(\mathbf{k}, \mathbf{k}, \omega) = \tilde{x}_{\mathbf{k}}^2 \left[ u_{\mathbf{k}}^2 \delta(E_{\mathbf{k}} - \omega) + v_{\mathbf{k}}^2 \delta(-E_{\mathbf{k}} - \omega) \right]
$$
$$
+ \frac{\tilde{y}_{\mathbf{k}}^2}{N^2} \sum_{\mathbf{q}(\neq \mathbf{k})} \left[ S(S+1)(\tilde{J}_{\mathbf{k}} + \tilde{J}_{\mathbf{q}})^2 + (\tilde{V}_{\mathbf{k}} + \tilde{V}_{\mathbf{q}})^2 \right] \left\{ \left[ \left( \frac{u_{\mathbf{k}} C_{1,\mathbf{kq}}^+}{E_{\mathbf{k}} - \tilde{E}_{\mathbf{q}}} \right)^2 + \left( \frac{v_{\mathbf{k}} C_{2,\mathbf{kq}}^-}{E_{\mathbf{k}} + \tilde{E}_{\mathbf{q}}} \right)^2 \right] \delta(\tilde{E}_{\mathbf{q}} + \omega) \right.
$$
$$
\left. + \left[ \left( \frac{u_{\mathbf{k}} C_{2,\mathbf{kq}}^-}{E_{\mathbf{k}} + \tilde{E}_{\mathbf{q}}} \right)^2 + \left( \frac{v_{\mathbf{k}} C_{1,\mathbf{kq}}^+}{E_{\mathbf{k}} - \tilde{E}_{\mathbf{q}}} \right)^2 \right] \delta(-\tilde{E}_{\mathbf{q}} + \omega) \right\} .
\tag{54}
$$

The first line describes the usual coherent dispersive excitation of a superconducting quasiparticle. The summation over $\mathbf{q}$ in the second and third line describes a continuum of excitations associated with scattering off the impurity, and the Kondo effect. It also contains the YSR bound state at $\mathbf{k}_0$, for which the coefficients $\tilde{J}_{\mathbf{q}}$ and $\tilde{V}_{\mathbf{q}}$ are singular (proportional $N$). In this case a dispersionless excitation appears at energies $\pm \tilde{E}_{\mathbf{k}_0}$. Moreover, the renormalized exchange energy $\tilde{J}_{\mathbf{q}}$ may become very large around the Fermi level due to the contributions from the expectation values in the term $A_{\mathbf{q}\lambda}^{(3)}$ in the renormalization equation (29). These contributions are assigned to the Kondo resonance [33].

## 3 Numerical results

We conclude by using the above-described self-consistent scheme to calculate a number of physical observables. The numerical calculation is carried out in the thermodynamic limit, $N \to \infty$. Thus, all summations over vectors $\mathbf{q} \neq \mathbf{k}_0$ are calculated by replacing the sum with an energy integral using the BCS density of states,

$$
\frac{1}{N} \sum_{\mathbf{q}(\neq \mathbf{k}_0)} \to \nu_0 \int_\Delta^W \Theta(|E| - \Delta) \frac{|E|}{\sqrt{E^2 - \Delta^2}} dE ,
\tag{55}
$$

where $\nu_0$ is the density of states in the normal state near the Fermi level and $W$ is the bandwidth of the conduction electron band. This description in the continuum limit is possible because for $\mathbf{q} \neq \mathbf{k}_0$ the quasiparticle energy remains unchanged. The introduced density of states fulfills the sum rule condition

$$
N = \nu_0 \int_\Delta^W \Theta(|E| - \Delta) \frac{|E|}{\sqrt{E^2 - \Delta^2}} dE .
\tag{56}
$$

The renormalization process starts with a guess for the expectation values $\langle \alpha_{\mathbf{k}}^\dagger \alpha_{\mathbf{k}'} \rangle$. Using these values, the bound state quantum number $\mathbf{k}_0$ and the corresponding energy $\tilde{E}_{\mathbf{k}_0}$ are found by evaluating Eq. (35). Next, the renormalization equations (29) and (30) as well as Eqs. (38), (44), and (46) are evaluated stepwise starting from $\lambda = 1$ down to $\lambda = 0$. The step width is chosen to be $\Delta\lambda = 10^{-3}$. The starting values are $E_{\mathbf{k},\lambda=1} = E_{\mathbf{k}}$, $x_{\mathbf{k},\lambda=1} = 1$, and $y_{\mathbf{k},\lambda=1} = 0$. In each $\lambda$-step, Eqs. (29) and (30) are solved self-consistently for $J_{\mathbf{k},\lambda}$ and $V_{\mathbf{k},\lambda}$, respectively. At the end of this procedure, we arrive with the fully renormalized values $\tilde{E}_{\mathbf{k}_0}$, $\tilde{J}_{\mathbf{k}}$, $\tilde{V}_{\mathbf{k}}$, $\tilde{x}_{\mathbf{k}}$, and $\tilde{y}_{\mathbf{k}}$. With all these quantities, we calculate the renormalized quasiparticle operator using Eq. (48), and, correspondingly, we are able to recalculate the expectation values $\langle \alpha_{\mathbf{k}}^\dagger \alpha_{\mathbf{k}'} \rangle$ using Eq. (41).

Next, we restart the entire renormalization process using these improved expectation values, and again solve all renormalization equations down to $\lambda = 0$. We conclude the iteration by recalculating the expectation values. This procedure is repeated until self-consistency is



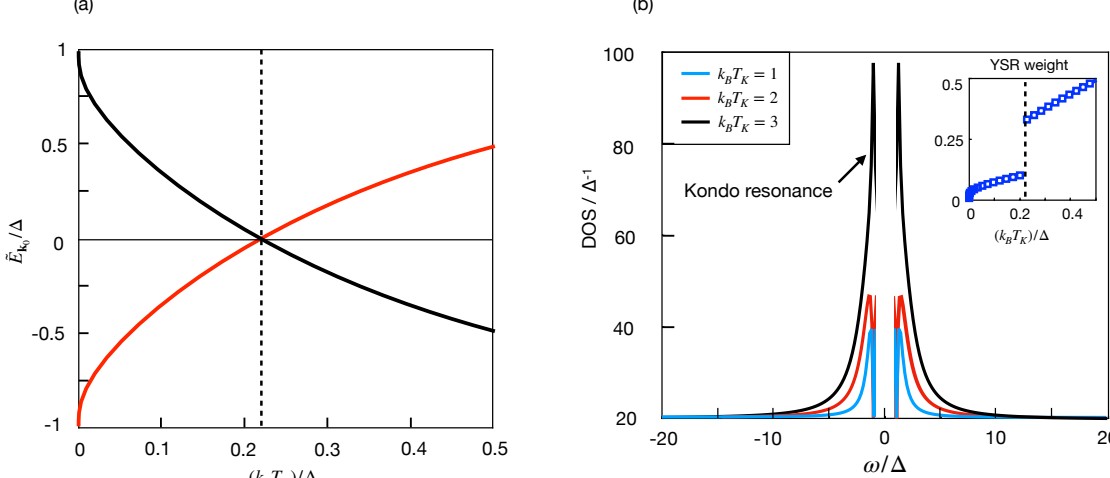

Figure 2: (a) Energy of the YSR state as a function of the Kondo temperature. (b) Calculated density of states at larger values of the Kondo temperature. A strong enhancement of the spectral intensity around the normal state Fermi level is found, which marks the crossover into a Kondo resonance regime. The inset shows the spectral weight of the YSR bound state around the values of $k_B T_K/\Delta$ where the bound state crossed zero energy. In agreement with literature results [28], a jump-like behaviour is found.

obtained. We define self-consistency to be fulfilled if the relative difference between the expectation values in two subsequent procedures is less than $10^{-3}$ for all $\mathbf{k}$.

For the evaluation of the spectral function which is given by Eq. (54) we describe the $\delta$-functions with Lorentzian functions with a manually-imposed broadening of $\eta = 0.03\Delta$. This value is taken in all figures showing spectral functions. Finally, all energy values in the figures are given in units of $\Delta$ unless it is stated otherwise.

## 3.1 YSR state energy and Kondo resonance

In Fig. 2(a), we show the result of the self-consistent solution for the energy $\tilde{E}_{\mathbf{k}_0}$ at $T = 0$ as a function of the Kondo temperature $k_B T_K$. Here, the Kondo temperature is varied by changing the exchange coupling parameter $J$. The relationship between $T_K$ and $J$ is

$$k_B T_K = W \sqrt{J \nu_0} \exp\left(-\frac{1}{J \nu_0}\right). \tag{57}$$

The energy of the second YSR excitation at $-\tilde{E}_{\mathbf{k}_0}$ is also shown (red line). One clearly recognizes the behavior known from former numerical studies [6, 28] including the crossing at $\tilde{E}_{\mathbf{k}_0} = 0$ (dashed line) which marks the transition from a weak coupling regime at low $T_K$ to a strong coupling regime at larger $T_K$ where the Kondo physics becomes significant. This regime is shown in Fig. 2(b) where the integrated spectral function from Eq. (54) is shown for three larger values of the Kondo temperature. It is seen that the spectral intensity is strongly enhanced around the superconducting gap edge. The width of this resonance peak scales with the Kondo temperature, which is a typical behavior of the Kondo resonance. The YSR states are pinned to the gap edges since the exchange coupling is already relatively large. In agreement with early numerical studies [28], our calculations also show that the spectral weight of the YSR state within the gap (shown in the inset) also increases with the Kondo temperature, and exhibits a characteristic jump when crossing zero energy.

## 3.2 Spectral function

To illustrate the behavior of the YSR state in the spectral function in more detail, we considered three further values of $J/\Delta$ corresponding to a rather weak coupling in the YSR regime, $J/\Delta = 20$ ($k_B T_K \approx 0.003$), an intermediate coupling $J/\Delta = 55$ ($k_B T_K \approx 0.120$), and a large value $J/\Delta = 120$ ($k_B T_K \approx 0.476$) in the Kondo regime.

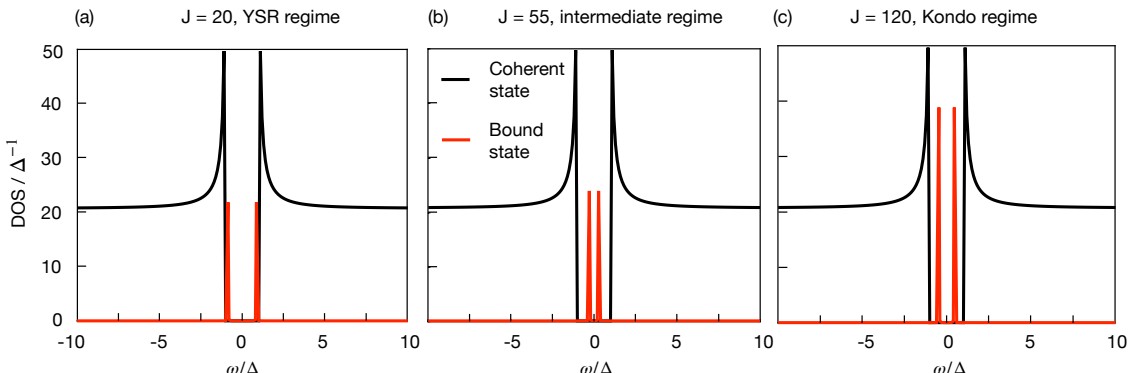

Figure 3: Total density of states for three different values of $J$ in an energy region around the Fermi level. These results where obtained by **k**-summation of both parts $\mathbf{k} = \mathbf{k}_0$ and $\mathbf{k} \neq \mathbf{k}_0$ of the spectral function. The superconducting gap is clearly seen. The bound state inside the gap is shown in a different color.

First, we turn to the **k**-integrated spectral function including the bound state quantum number $\mathbf{k}_0$. Fig. 3 shows the spectral density in a narrow region around the Fermi level. The superconducting gap and the density of states of the conduction electrons are not found to vary noticeably with $J$. The excitation peak of the YSR state (shown in red color), on the other hand, moves in energy as $J$ changes. Furthermore, the spectral intensity of the YSR state changes, in particular, once the Kondo regime is entered (compare also the inset in Fig. 2(b)). As a crosscheck, we verified that the energy integral over the total density of states yields the total number of states $N$, where in the continuum limit considered here $N$ is given by the integral over the density of states in Eq. (56).

More interesting is the **k**-resolved spectral function, which for $\mathbf{k} \neq \mathbf{k}_0$ is shown in Fig. 4. In the weak coupling YSR regime at $J/\Delta = 20$, the spectrum is composed of a dispersive excitation and a YSR bound state located close to the gap edge. The dispersive excitation loses spectral weight once the peak approaches $k_F$, while the spectral weight of the YSR state increases.

A larger value of $J$ is considered in Fig. 4(b). The two YSR peaks move closer together, and become somewhat more intense. At the same time, the coherent excitation weakens. The YSR state now also has a somewhat larger extent in **k**, but still remains relatively close to $k_F$.

If, finally, the exchange coupling is increased even more, the YSR states cross zero energy. As shown in Fig. 4(c), the bound state spectral weight coming from momenta $\mathbf{k} \neq \mathbf{k}_0$ is now much larger than that in the weak coupling YSR regime.

### 3.3 Influence of temperature and spin size

To further benchmark our approach against purely numerical methods, we now turn to the dependence of the bound state on temperature and spin size. Fig. 5 shows results of the YSR bound state energy and weight as a function of the temperature $T$ and the size $S$ of the local spin. Panel (a) shows the influence of the temperature $T$ in the two different regimes at $J/\Delta = 20$ and $J/\Delta = 120$. In agreement with the numerical results from Ref. [6], we

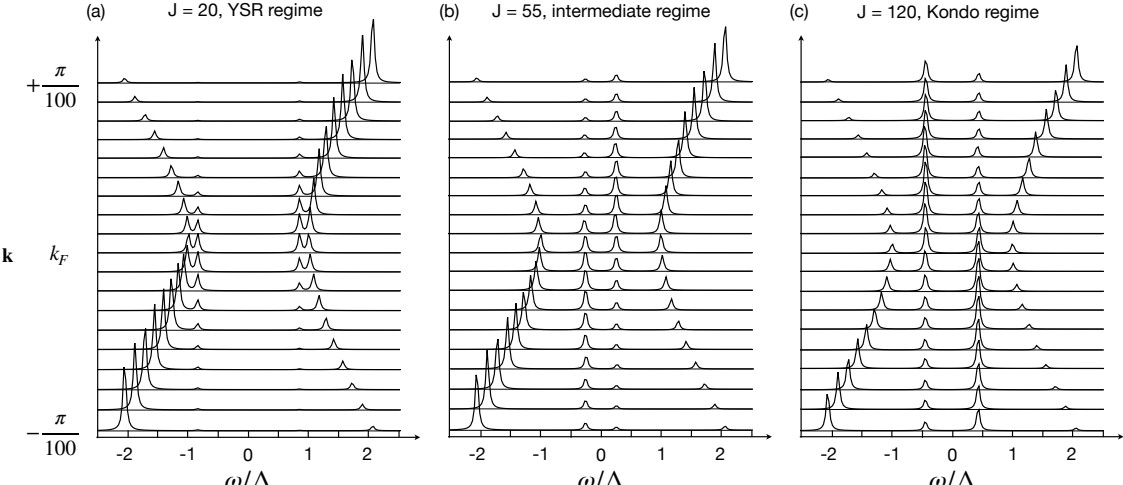

Figure 4: Spectral functions for the same parameters as in Fig. 3 but only for $\mathbf{k} \neq \mathbf{k}_0$. The bound state at $\tilde{E}_{\mathbf{k}_0}$ becomes extended in $\mathbf{k}$, and the distribution of the spectral weight gets inverted when the Kondo regime is entered.

find that the YSR bound state energy is shifted to smaller values at small exchange couplings. This effect becomes inverted in the Kondo regime, where the shift is smaller but the energy increases with temperature. This is seen even more clearly in the panels (b,d) where we have directly calculated the $\mathbf{k}$-integrated spectral function in a region inside the superconducting gap. If we compare the zero temperature result (blue line) with the one at the somewhat higher temperature $k_B T = 0.35$ below $T_c$, we see that the pair of YSR peaks comes closer together at higher temperature in the weak coupling case, while the opposite is the case at strong coupling. This is exactly the behavior found in Ref. [6]. Even the slight change of the intensity with respect to temperature agrees with the mentioned numerical study.

Fig. 5(c) finally depicts the dependence of the YSR state energy on the size of the quantum spin at fixed $T$ (below the critical temperature $T_c$). We observe a strong reduction of $\tilde{E}_{\mathbf{k}_0}$ with $S$ in all regimes. As expected, the most significant effect of quantum fluctuations is obtained at small values of $S$ where the energy $\tilde{E}_{\mathbf{k}_0}$ deviates most strongly from the classical spin limit (dashed lines). In the YSR regime, however, the effect of the quantum nature of the spin is indeed rather weak, but it extends over a wider range of $S$ than in the Kondo regime.

## 4 Conclusion

In this manuscript, we have presented a new approach to solve the paradigmatic Hamiltonian describing a local quantum spin coupled to a superconducting substrate. The basic idea of our approach is to eliminate the interaction between electrons and the impurity by unitary transformations that yields a diagonal renormalized Hamiltonian. In this process, bound states manifest themselves as singularities in the unitary transformation. Postponing a detailed derivation to a subsequent publication, we stress that our method can straightforwardly be extended to larger numbers of impurities. This has been worked out already for the Kondo lattice model in the normal state [33].

We first illustrated how the spin-fermion exchange interaction is integrated out using a stepwise unitary transformation. Performing the transformation in small steps allowed us consider the lowest order commutators in a well-controlled way. We have derived renormalization equations for the quasiparticle energy as well as operators. We have shown that our method

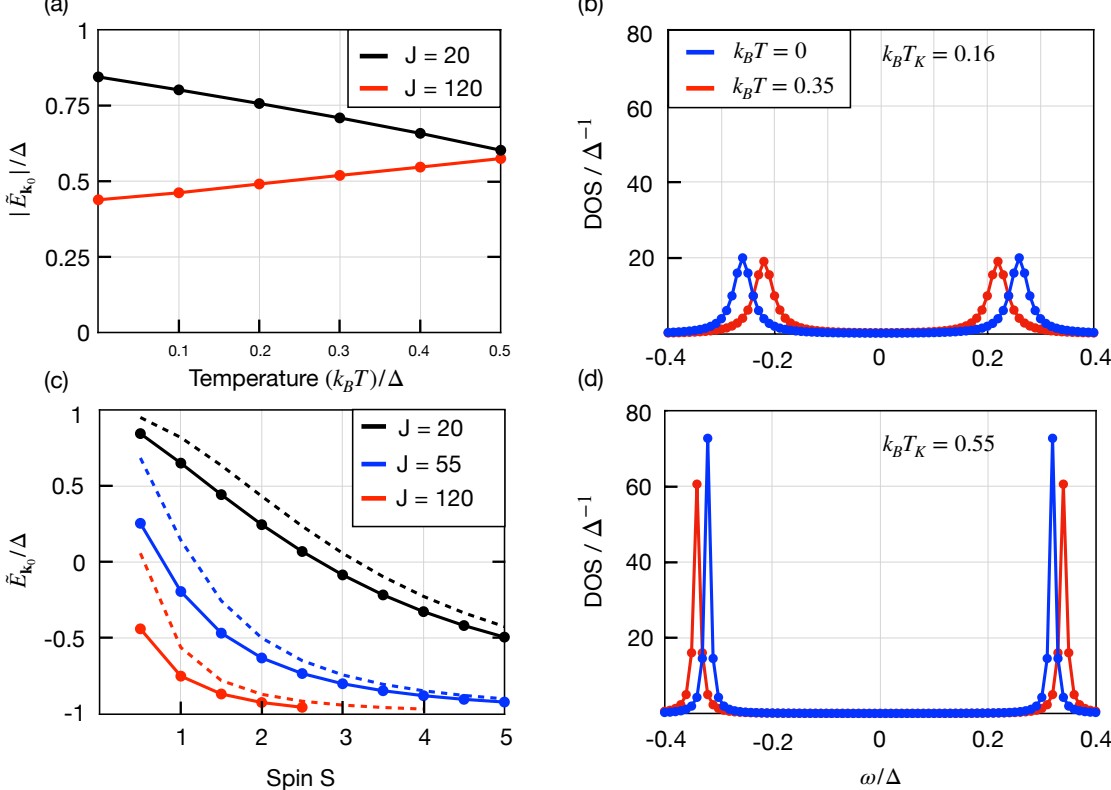

Figure 5: Dependence of the YSR bound state energy of a quantum spin on temperature and spin size. (a) Variation of the temperature. In agreement with former numerical studies [6], the YSR state energy decreases with temperature while in the Kondo regime it slightly increases. (b,d) **k**-integrated spectral function showing the pair of YSR bound states in a narrow energy region inside the superconducting gap. Shown are the spectra at two temperatures (blue and red line) and two different Kondo temperatures (upper/lower panel). Again a very good agreement with Ref. [6] is found. (c) Variation of the spin for the three different values of $J$ discussed above. The corresponding results for the classical spin are also shown (dashed lines). $\tilde{E}_{\mathbf{k}_0}$ approaches the value of the classical spin case in the limit of large $S$ in all cases.

fully captures Yu-Shiba-Rusinov bound states and the Kondo resonance within one uniform theoretical framework. While the YSR state is a $\delta$ peak inside the superconducting gap, the Kondo resonance appears outside the gap as a peak of finite width which is proportional to the Kondo temperature. Our approach hence draws a clear picture of the physics in the superconducting Kondo model by capturing on equal footing both the classical scattering problem of a magnetic impurity, and also the complex many-body effect leading to the Kondo resonance. In addition, we showed that our approach also yields access to effectively momentum-resolved observables, which for example are important in the context of ARPES-experiments.

In contrast to purely numerical approaches, our method also allows for detailed analytical insights. This power of our approach is rooted in the fact that bound states manifest themselves as singularities, and that it is therefore sufficient to extract this singular behavior from the full picture to investigate bound state physics. This for example allows us to explicitly construct bound state quasiparticle operators, which in turn reveals how increasing quantum fluctuations of the local spin pushes the bound state from a YSR-like form to a Kondo-like expression.

## Acknowledgements

We would like to thank K.W. Becker for helpful discussions.

**Funding information**   This project has received funding from the Deutsche Forschungsgemeinschaft via the Emmy Noether Programme ME4844/1-1 (project id 327807255), the Collaborative Research Center SFB 1143 (project id 247310070), and the Cluster of Excellence on Complexity and Topology in Quantum Matter ct.qmat (EXC 2147, project id 390858490).

## A   Classical spin

Here we show that the specific case of a classical spin leads to an exactly solvable scattering problem which gives rise to the well known YSR bound states in the absence of the Kondo resonance. To describe a classical spin, the spin operator $\mathbf{S}$ is replaced by a constant vector. We choose the spin to be oriented along $z$ and thus set $\mathbf{S} = S\,\mathbf{e}_z$ ($\mathbf{e}_z$ being the unit vector along $z$). The electron-impurity Hamiltonian then takes the form

$$\mathcal{H}_{1,cl} = \frac{JS}{2N} \sum_{\mathbf{kk'}} \left[ (u_\mathbf{k} v_\mathbf{k'} - u_\mathbf{k'} v_\mathbf{k}) \left( \alpha_\mathbf{k}^\dagger \beta_\mathbf{k'}^\dagger + \beta_\mathbf{k'} \alpha_\mathbf{k} \right) + (u_\mathbf{k} u_\mathbf{k'} + v_\mathbf{k'} v_\mathbf{k}) \left( \alpha_\mathbf{k}^\dagger \alpha_\mathbf{k'} - \beta_\mathbf{k}^\dagger \beta_\mathbf{k'} \right) \right]. \tag{58}$$

Because a classical spin has no quantum fluctuations, the Hamiltonian is entirely quadratic in operators.

Applying our approach described above to the scattering problem of the classical spin simply means that the last part in Eq. (28) is omitted since it arises from the commutator between spin operators, which is zero in the classical spin case. Furthermore, the number $S(S+1)$ is replaced with $S^2$ according to the convention in Eq. (58). Thus, due to $A_{\mathbf{k},\lambda}^{(3)} = 0$ the renormalization equations (29) and (30) simplify to

$$-\frac{J}{2} = -J_{\mathbf{k},\lambda} + \frac{J\lambda}{2} \frac{1}{N} \sum_{\mathbf{q}(\neq\mathbf{k})} \frac{V_{\mathbf{k},\lambda} + V_{\mathbf{q},\lambda}}{E_{\mathbf{q},\lambda}^2 - E_{\mathbf{k},\lambda}^2} \left( E_{\mathbf{k},\lambda} + \frac{\Delta - \varepsilon_\mathbf{q}}{E_\mathbf{q}} E_{\mathbf{q},\lambda} \right), \tag{59}$$

$$0 = V_{\mathbf{k},\lambda} - \frac{J\lambda}{2} S^2 \frac{1}{N} \sum_{\mathbf{q}(\neq\mathbf{k})} \frac{J_{\mathbf{k},\lambda} + J_{\mathbf{q},\lambda}}{E_{\mathbf{q},\lambda}^2 - E_{\mathbf{k},\lambda}^2} \left( E_{\mathbf{k},\lambda} + \frac{\Delta - \varepsilon_\mathbf{q}}{E_\mathbf{q}} E_{\mathbf{q},\lambda} \right). \tag{60}$$

The renormalization equations (32) and (33) for the YSR bound state read for the classical spin case

$$-\frac{J}{2} = -J_{\mathbf{k}_0,\lambda} + \frac{J\lambda}{2} \frac{V_{\mathbf{k}_0,\lambda}}{N} \sum_{\mathbf{q}(\neq\mathbf{k}_0)} \frac{E_{\mathbf{k}_0,\lambda} + \Delta - \varepsilon_\mathbf{q}}{E_\mathbf{q}^2 - E_{\mathbf{k}_0,\lambda}^2}, \tag{61}$$

$$V_{\mathbf{k}_0,\lambda} = -\frac{J\lambda}{2} S^2 \frac{J_{\mathbf{k}_0,\lambda}}{N} \sum_{\mathbf{q}(\neq\mathbf{k}_0)} \frac{E_{\mathbf{k}_0,\lambda} + \Delta - \varepsilon_\mathbf{q}}{E_\mathbf{q}^2 - E_{\mathbf{k}_0,\lambda}^2}. \tag{62}$$

Since we consider the renormalization in the very first renormalization step we can set the $\lambda$ factor equal to 1. Then, combining the two equations by elimination of $V_{\mathbf{k}_0,\lambda}$ and solving the resulting equation for $J_{\mathbf{k}_0,\lambda}$ we obtain

$$J_{\mathbf{k}_0,\lambda} = \frac{J/2}{1 - \left( \frac{J}{2} \frac{S}{N} \sum_{\mathbf{q}(\neq\mathbf{k}_0)} \frac{E_{\mathbf{k}_0,\lambda} + \Delta - \varepsilon_\mathbf{q}}{E_\mathbf{q}^2 - E_{\mathbf{k}_0,\lambda}^2} \right)^2}. \tag{63}$$

It is clearly seen that a singularity of $J_{\mathbf{k}_0,\lambda}$, which is the condition for a renormalization of $E_{\mathbf{k}_0,\lambda}$, is found if the equation

$$1 = \frac{J}{2}\frac{S}{N}\sum_{\mathbf{q}(\neq\mathbf{k}_0)}\frac{E_{\mathbf{k}_0,\lambda}+\Delta-\varepsilon_{\mathbf{q}}}{E_{\mathbf{q}}^2-E_{\mathbf{k}_0,\lambda}^2} \tag{64}$$

is fulfilled. The solution $E_{\mathbf{k}_0,\lambda}$ of this equation is the renormalized energy value due to the singularity. To explicitly calculate $E_{\mathbf{k}_0,\lambda}$, we use the BCS density of states,

$$\nu(E) = \nu_0\Theta(|E|-\Delta)\frac{|E|}{\sqrt{E^2-\Delta^2}}\,, \tag{65}$$

where $\nu_0$ is the density of states at the Fermi level in the normal state. We also express the summation in terms of an energy integration and calculate the corresponding integral analytically. The result is

$$\frac{1}{N}\sum_{\mathbf{q}(\neq\mathbf{k}_0)}\frac{E_{\mathbf{k}_0,\lambda}+\Delta-\varepsilon_{\mathbf{q}}}{E_{\mathbf{q}}^2-E_{\mathbf{k}_0,\lambda}^2} = \nu_0\pi\frac{\Delta+E_{\mathbf{k}_0,\lambda}}{\sqrt{\Delta^2-E_{\mathbf{k}_0,\lambda}^2}}\,. \tag{66}$$

Thus, the renormalized energy $E_{\mathbf{k}_0,\lambda}$ is given by the solution of the equation

$$1 - \frac{JS}{2}\nu_0\pi\frac{\Delta+E_{\mathbf{k}_0,\lambda}}{\sqrt{\Delta^2-E_{\mathbf{k}_0,\lambda}^2}} = 0\,. \tag{67}$$

This equation has solutions for $E_{\mathbf{k}_0,\lambda}$ only in the superconducting state, i. e. if $\Delta\neq 0$. As described in Sec. 2.2 there are two solutions of Eq. (67) where one lies in the conduction band, $E_{\mathbf{k}_0}>\Delta$, which is the starting value of the bound state renormalization, and the other one, $E_{\mathbf{k}_0,\lambda}<\Delta$, is inside the gap. The latter reads

$$E_{\mathbf{k}_0,\lambda} = \tilde{E}_{\mathbf{k}_0} = \Delta\frac{1-\left(\frac{JS\nu_0\pi}{2}\right)^2}{1+\left(\frac{JS\nu_0\pi}{2}\right)^2}\,, \tag{68}$$

which is the final value of the renormalization and therefore it corresponds to the bound state energy. This solution exactly agrees with the original result obtained by Shiba [2]. Thus, the particular state $\mathbf{k}_0$ where the singularity appears has to be identified with the well-known YSR state appearing in the spectral function as a pair of peaks inside the superconducting gap.

In summary, the whole renormalization process occurs as follows. At the starting point $\lambda=1$ of the renormalization process the particular state $\mathbf{k}_0$ where $E_{\mathbf{k}_0}=\Delta$ becomes a singular point for the transformation coefficient $J_{\mathbf{k}_0,\lambda}$ (and also $V_{\mathbf{k}_0,\lambda}$), i. e. it takes a value proportional to $N$. Thus, the solution for the bound state is described by $\tilde{J}_{\mathbf{k}_0} = N\tilde{J}$ and $\tilde{V}_{\mathbf{k}_0} = -S\tilde{J}_{\mathbf{k}_0}$ due to Eqs. (62) and (64). The corresponding equation for $\tilde{J}$ is found from Eq. (31) by taking into account again Eq. (64),

$$\tilde{J} = \frac{\Delta-\tilde{E}_{\mathbf{k}_0}}{2S}\,. \tag{69}$$

Using this expression and Eq. (48) we obtain the explicit form of the quasiparticle operator with respect to $\mathbf{k}\neq\mathbf{k}_0$ in the classical spin case,

$$\tilde{\alpha}_{\mathbf{k}}^\dagger = \tilde{x}_{\mathbf{k}}\alpha_{\mathbf{k}}^\dagger + \frac{\tilde{y}_{\mathbf{k}}}{N}\sum_{\mathbf{k}'(\neq\mathbf{k},\mathbf{k}_0)}\left[C_{1,\mathbf{k}'\mathbf{k}}^+\frac{(\tilde{J}_{\mathbf{k}}+\tilde{J}_{\mathbf{k}'})S-(\tilde{V}_{\mathbf{k}}+\tilde{V}_{\mathbf{k}'})}{E_{\mathbf{k}'}-E_{\mathbf{k}}}\alpha_{\mathbf{k}'}^\dagger - C_{2,\mathbf{k}\mathbf{k}'}^-\frac{(\tilde{J}_{\mathbf{k}}+\tilde{J}_{\mathbf{k}'})S+(\tilde{V}_{\mathbf{k}}+\tilde{V}_{\mathbf{k}'})}{E_{\mathbf{k}}+E_{\mathbf{k}'}}\beta_{\mathbf{k}'}\right]$$

$$+ (u_{\mathbf{k}}+v_{\mathbf{k}})\frac{\Delta-\tilde{E}_{\mathbf{k}_0}}{E_{\mathbf{k}}-\tilde{E}_{\mathbf{k}_0}}\alpha_{\mathbf{k}_0}^\dagger\,. \tag{70}$$

It is seen that the quasiparticle in the classical spin case consists of three basic contributions. The first part proportional to $\tilde{x}_{\mathbf{k}}$ describes the coherent excitation which follows the original dispersion of the superconducting quasiparticle. The second part in the first line which is proportional to $\tilde{y}_{\mathbf{k}}$ contains the scattering to all other states except $\mathbf{k}$ itself and $\mathbf{k}_0$. Since in contrast to the quantum spin case the renormalization of the exchange coupling, $\tilde{J}_{\mathbf{k}}$, is always very small (absence of the Kondo resonance) this term only produces a very small spectral weight ($\propto 1/N$) in the single-particle spectral function. Finally, the third term in the second line is the creation of the YSR bound state $\mathbf{k}_0$ appearing at an energy $\tilde{E}_{\mathbf{k}_0}$ inside the superconducting gap. Thus, the expression for the single-particle spectral function for $\mathbf{k} \neq \mathbf{k}_0$ given by Eq. (54) simplifies in the classical spin case to

$$
\Im G(\mathbf{k}, \mathbf{k}, \omega) = \frac{\tilde{x}_{\mathbf{k}}^2}{2} \left[ \left( 1 + \frac{\varepsilon_{\mathbf{k}}}{E_{\mathbf{k}}} \right) \delta(E_{\mathbf{k}} - \omega) + \left( 1 - \frac{\varepsilon_{\mathbf{k}}}{E_{\mathbf{k}}} \right) \delta(-E_{\mathbf{k}} - \omega) \right]
$$
$$
+ \frac{1}{2} \left( \frac{\Delta - \tilde{E}_{\mathbf{k}_0}}{E_{\mathbf{k}} - \tilde{E}_{\mathbf{k}_0}} \right)^2 \left[ \delta(\tilde{E}_{\mathbf{k}_0} - \omega) + \delta(-\tilde{E}_{\mathbf{k}_0} - \omega) \right]. \tag{71}
$$

In addition to the usual dispersive states (first line), there appears in the second line a pair of dispersionless excitations which are attributed to the YSR bound states. Those have the $\mathbf{k}$-independent energy $\pm \tilde{E}_{\mathbf{k}_0}$ above and below the Fermi level, but inside the superconducting gap. Since all excitations appear as $\delta$ peaks without broadening the Kondo resonance arising from many-particle excitations is absent here.

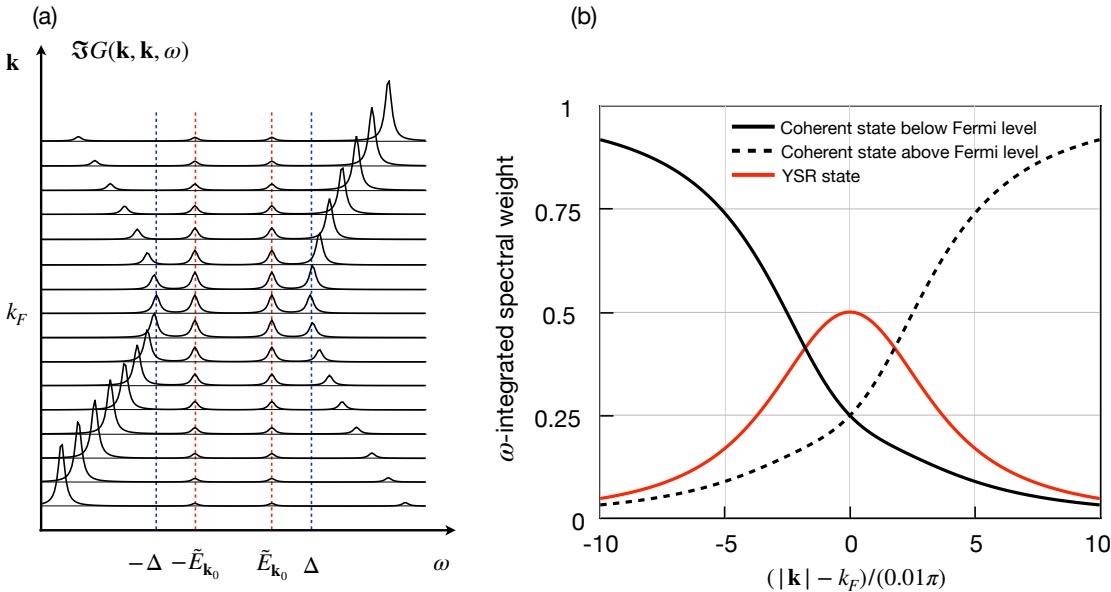

Figure 6: (a) Single-particle Green's function for a classical spin for $\mathbf{k} \neq \mathbf{k}_0$ near the Fermi surface. The chosen parameters are $J = \Delta = 0.004W$, $\tilde{E}_{\mathbf{k}_0} = 0.002W$, and $S = 1/2$ ($W$: band width). In this figure the level broadening was fixed manually to the value $\eta = 2 \cdot 10^{-4}W$. (b) Energy-integrated spectral weight of the different excitations in the spectral function of panel (a) as a function of $\mathbf{k}$.

The behavior of the YSR state in the spectral function is visualized in Fig. 6(a) where the spectral function from Eq. (71) is plotted explicitly for a specific set of parameters, and for a number of momenta $\mathbf{k} \neq \mathbf{k}_0$. One finds the typical dispersionless double-peak structure of the YSR state inside the superconducting gap where the intensity is strongest in the vicinity of the

gap edge. Furthermore, one clearly recognizes the two distinct branches of the superconducting quasiparticles below and above the Fermi level. The spectral weight of the corresponding excitations which are shown in Fig. 6(b) reveals an intensity maximum of the YSR state near the Fermi momentum of the normal state. The **k** dependence of the coherent excitation intensity, however, is mainly influenced by the Bugoliubov coefficients $|u_{\mathbf{k}}|^2$ and $|v_{\mathbf{k}}|^2$.

## B  Non-magnetic impurity

Here we explain our approach to bound states by applying it to the perhaps most simple case of a local potential impurity embedded into a spinless, normal-state electron system. Concretely, we consider a toy-model Hamiltonian $\mathcal{H}_V$ composed of an electron-part $\mathcal{H}_0$ and the electron-impurity-part $\mathcal{H}_{1,V}$ proportional to the coupling strength $V$,

$$\mathcal{H}_V = \mathcal{H}_0 + \mathcal{H}_{1,V}, \quad \text{with} \quad \mathcal{H}_0 = \sum_k \varepsilon_k c_k^\dagger c_k \quad \text{and} \quad \mathcal{H}_{1,V} = \frac{V}{2N} \sum_{k \neq k'} \left( c_k^\dagger c_{k'} + c_{k'}^\dagger c_k \right). \quad (72)$$

Here, $\varepsilon_k$ is the bare electronic dispersion, which is a function of the one-dimensional momentum $k$, and the lattice consists of $N$ sites.

Following the idea of Sec. 2.1, we subject the Hamiltonian (72) to an exact mapping that brings the Hamiltonian to a new form $\tilde{\mathcal{H}}_V$. The mapping between the original Hamiltonian $\mathcal{H}_V$ and $\tilde{\mathcal{H}}_V$ is implemented by a unitary transformation of the form given in Eq. (8). The transformed Hamiltonian should not only be quadratic in the original fermionic operators $c_k^\dagger$, but also have the diagonal form

$$\tilde{\mathcal{H}}_V = \sum_k \tilde{\varepsilon}_k c_k^\dagger c_k, \quad (73)$$

with a renormalized dispersion $\tilde{\varepsilon}_k$. This treatment is technically slightly different (but equivalent in the way explained in the main text) from the usual diagonalization in which the quasiparticle operators in the Hamiltonian are rotated in the operator subspace. For technical convenience, our approach keeps the single-particle operators in their original basis while the Hamiltonian is renormalized. Therefore, $\tilde{\mathcal{H}}_V$ differs in its form from $\mathcal{H}_V$, but has the same eigenvalue spectrum as the original model.

As explained above in the context of the spin impurity the unitary transformation to obtain $\tilde{\mathcal{H}}_V$ is not carried out in one single step but within a process of steps which are described by a parameter $\lambda$. The $\lambda$-dependent Hamiltonian for the present problem is defined as follows

$$\mathcal{H}_{V,\lambda} = \mathcal{H}_{0V,\lambda} + \mathcal{H}_{1V,\lambda}, \quad \text{with} \quad \mathcal{H}_{0V,\lambda} = \sum_k \varepsilon_{k,\lambda} c_k^\dagger c_k, \quad \mathcal{H}_{1V,\lambda} = \lambda \frac{V}{2N} \sum_{k \neq k'} \left( c_k^\dagger c_{k'} + c_{k'}^\dagger c_k \right). \quad (74)$$

The unitary transformation combining two subsequent parameter values $\lambda$ and $\lambda - \Delta\lambda$ is constructed such that applying the transformation as in Eq. (11) the structure of $\mathcal{H}_{V,\lambda}$ is kept for any value of $\lambda$. A convenient ansatz for $X_{V,\lambda,\Delta\lambda}$ is

$$X_{V,\lambda,\Delta\lambda} = \frac{\Delta\lambda}{N} \sum_{k \neq k'} \frac{A_{k,\lambda}}{\varepsilon_{k,\lambda} - \varepsilon_{k',\lambda}} \left( c_k^\dagger c_{k'} - c_{k'}^\dagger c_k \right). \quad (75)$$

The real coefficients $A_{k,\lambda}$ will again turn out to be key in the description of bound states. To determine their renormalization equations, we plug the ansatz (75) into Eq. (11), which allows to determine equations for $A_{k,\lambda}$ from an expansion as in Eq. (13) in terms of commutators up to the lowest order in $\Delta\lambda$. Up to first order in the generator $X_V$, the commutators of $X_V$ with

$\mathcal{H}_0$ and $\mathcal{H}_{1,V}$ as defined in Eq. (72) read,

$$[X_{V,\lambda,\Delta\lambda}, \mathcal{H}_{0V,\lambda}] = -\frac{\Delta\lambda}{N} \sum_{k \neq k'} A_{k,\lambda} \left( c_k^\dagger c_{k'} + c_{k'}^\dagger c_k \right),$$

$$[X_{V,\lambda,\Delta\lambda}, \mathcal{H}_{1V,\lambda}] = \frac{V}{N^2} \lambda \Delta\lambda \sum_{kk'} \sum_{q(\neq k)} \frac{A_{k,\lambda} + A_{q,\lambda}}{\varepsilon_{k,\lambda} - \varepsilon_{q,\lambda}} \left( c_k^\dagger c_{k'} + c_{k'}^\dagger c_k \right).$$

(76)

Comparing again the both sides of Eq. (13) after plugging in the two above commutators we find the following renormalization equations,

$$-\frac{V}{2} = -A_{k,\lambda} + \frac{V}{N} \lambda \sum_{q(\neq k)} \frac{A_{k,\lambda} + A_{q,\lambda}}{\varepsilon_{k,\lambda} - \varepsilon_{q,\lambda}},$$

(77)

$$\varepsilon_{k,\lambda-\Delta\lambda} = \varepsilon_{k,\lambda} + \frac{2V}{N^2} \lambda \Delta\lambda \sum_{q(\neq k)} \frac{A_{k,\lambda} + A_{q,\lambda}}{\varepsilon_{k,\lambda} - \varepsilon_{q,\lambda}}.$$

(78)

Again $A_{k,\lambda}$ can become singular at a specific value of $k = k_0$, which indicates the formation of a bound state. Furthermore, since $A_{k_0,\lambda} \propto N$ such a singularity entails a renormalization of the single-particle energy from the bare value $\varepsilon_{k_0}$ to a new value – but *only* at the $k_0$ for which $A_{k_0,\lambda}$ is singular, i. e. for all $q \neq k_0$ we can set $\varepsilon_{q,\lambda} = \varepsilon_q$. The renormalization at $k_0$ will again be related to the binding energy of the bound state. Thus, for the singular state $k_0$ we can solve Eq. (77) for $A_{k_0,\lambda}$,

$$A_{k_0,\lambda} = \frac{V/2}{1 - \lambda \frac{V}{N} \sum_{q(\neq k_0)} \frac{1}{\varepsilon_{k_0,\lambda} - \varepsilon_q}}.$$

(79)

In the beginning of the renormalization process we can set $\lambda = 1$. Thus, a singularity is found if the equation

$$1 = \frac{V}{N} \sum_{q(\neq k_0)} \frac{1}{\varepsilon_{k_0,\lambda=1} - \varepsilon_q}$$

(80)

is fulfilled. The solution $\varepsilon_{k_0,\lambda=1}$ of this equation is the renormalized energy value due to the singularity. In general one finds two solutions, where the first solution $\varepsilon_{k_0}$ in general is a part of the quasiparticle band and the second solution lies below or above this band depending on the sign of $V$. It follows that the first solution $\varepsilon_{k_0}$ changes to the renormalized value $\varepsilon_{k_0,\lambda=1}$ and remains at this value down to $\lambda = 0$, i. e. $\varepsilon_{k_0,\lambda=1} = \tilde{\varepsilon}_{k_0}$. All other $q \neq k_0$ are unaffected and the corresponding energies remain unchanged, $\varepsilon_{q,\lambda} = \varepsilon_q$.

The fully renormalized energy value $\tilde{\varepsilon}_{k_0}$ which is a solution of Eq. (80) exactly agrees with the result of the well known t-matrix approach to bound states [36]. The bound state energy is calculated by evaluating the sum using the density of states of the free system and solving the resulting equation for $\tilde{\varepsilon}_{k_0}$. Usually the energy value $\tilde{\varepsilon}_{k_0}$ lies below the bottom of the conduction electron band and it only exists if $V < 0$, i. e. in the case of an attractive scattering potential.

The whole renormalization process occurs similar to the spin impurity problem as follows. At the starting point $\lambda = 1$ of the renormalization process a particular state $k_0$ from the quasiparticle band becomes a singular point for the corresponding transformation coefficient $A_{k_0,\lambda}$, i. e. it takes a value proportional to $N$. Thus, the solution for the coefficient at the bound state can be written in the form $\tilde{A}_{k_0} = N\tilde{A}$. The corresponding equation for $\tilde{A}$ is found from Eq. (78),

$$\tilde{A} = \frac{\tilde{\varepsilon}_{k_0} - \varepsilon_{k_0}}{2}.$$

(81)

For all other $k \neq k_0$, the coefficients $A_{k,\lambda}$ are found by self consistent solution of Eq. (77).

Let us again visualize the bound state by studying the spectral function, which is defined via the imaginary part of the single-particle Green's function,

$$\Im G(k,k,\omega) = \frac{1}{2\pi} \int_{-\infty}^{\infty} \langle [c_k^\dagger(-t), c_k]_+ \rangle e^{i\omega t} dt. \tag{82}$$

The time dependence and the expectation value are now formed with the Hamiltonian $\mathcal{H}_V$. Correspondingly, the expectation value of an operator $\mathcal{A}$ is defined as in Eq. (39) but now formed with $\mathcal{H}_V$. Using the Mori-Zwanzig formalism [37,38], the anticommutator correlation function can be rewritten to the form

$$\Im G(k,k,\omega) = \langle [c_k, \delta(\mathbf{L}_V - \omega)c_k^\dagger]_+ \rangle, \tag{83}$$

where $\mathbf{L}_V \mathcal{A} = [\mathcal{H}_V, \mathcal{A}]$ is the Liouville operator. Our method to calculate this quantity is again by using the invariance of the trace under a unitary transformation. This yields

$$\Im G(k,k,\omega) = \langle [\tilde{c}_k, \delta(\tilde{\mathbf{L}}_V - \omega)\tilde{c}_k^\dagger]_+ \rangle. \tag{84}$$

To find the transformed operators $\tilde{c}_k$, we use the same procedure that we used above to determine the fully renormalized Hamiltonian $\tilde{\mathcal{H}}_V$ from the original Hamiltonian. We start with an ansatz for the renormalized single-particle operator where the transformation is already carried out until an arbitrary $\lambda$. This ansatz can be suggested by forming the first order commutator of the single-particle operator and $X_{V,\lambda,\Delta\lambda}$ where we use the same operator form but with $\lambda$-dependent prefactors,

$$c_{k,\lambda}^\dagger = x_{k,\lambda} c_k^\dagger + y_{k,\lambda} \frac{1}{N} \sum_{q(\neq k)} \frac{A_{k,\lambda} + A_{q,\lambda}}{\varepsilon_{q,\lambda} - \varepsilon_{k,\lambda}} c_q^\dagger, \tag{85}$$

where the initial conditions at $\lambda = 1$ are $x_{k,\lambda=1} = 1$ and $y_{k,\lambda=1} = 0$. Now we find the renormalization equations for the $\lambda$-dependent coefficients $x_{k,\lambda}$ and $y_{k,\lambda}$ in Eq. (85) by evaluating a similar equation as Eq. (11) but for the renormalized single particle operator,

$$c_{k,\lambda-\Delta\lambda}^\dagger = e^{X_{V,\lambda,\Delta\lambda}} c_{k,\lambda}^\dagger e^{-X_{V,\lambda,\Delta\lambda}}, \tag{86}$$

where the operator expression as given by Eq. (75) must be used for $X_{V,\lambda,\Delta\lambda}$. Evaluating again the right hand side up to the linear order in $\Delta\lambda$ and using the above ansatz for $c_{k,\lambda}^\dagger$ we find the following set of renormalization equations for the $\lambda$-dependent coefficients,

$$x_{k,\lambda-\Delta\lambda} = x_{k,\lambda} - y_{k,\lambda} \frac{\Delta\lambda}{N^2} \sum_{q(\neq k)} \left( \frac{A_{k,\lambda} + A_{q,\lambda}}{\varepsilon_{q,\lambda} - \varepsilon_{k,\lambda}} \right)^2,$$

$$1 = x_{k,\lambda-\Delta\lambda}^2 + y_{k,\lambda-\Delta\lambda}^2 \frac{1}{N^2} \sum_{q(\neq k)} \left( \frac{A_{k,\lambda-\Delta\lambda} + A_{q,\lambda-\Delta\lambda}}{\varepsilon_{q,\lambda-\Delta\lambda} - \varepsilon_{k,\lambda-\Delta\lambda}} \right)^2. \tag{87}$$

This set of renormalization equations is solved simultaneously with the corresponding equations for the Hamiltonian until $\lambda = 0$ is obtained. Then the Hamiltonian is diagonal and the fully renormalized single-particle operator at $\lambda = 0$ is obtained from Eq. (85),

$$\tilde{c}_k^\dagger = \tilde{x}_k c_k^\dagger + \tilde{y}_k \frac{1}{N} \sum_{q(\neq k)} \frac{\tilde{A}_k + \tilde{A}_q}{\tilde{\varepsilon}_q - \tilde{\varepsilon}_k} c_q^\dagger. \tag{88}$$

Note that a possible bound state $k_0$ may be also included in the summation. If we consider a state $k \neq k_0$ we find

$$\tilde{c}_k^\dagger = \tilde{x}_k c_k^\dagger + \tilde{y}_k \frac{1}{N} \sum_{q(\neq k,k_0)} \frac{\tilde{A}_k + \tilde{A}_q}{\varepsilon_q - \varepsilon_k} c_q^\dagger + \frac{1}{2} \frac{\tilde{\varepsilon}_{k_0} - \varepsilon_{k_0}}{\tilde{\varepsilon}_{k_0} - \varepsilon_{k_0}} c_{k_0}^\dagger. \tag{89}$$

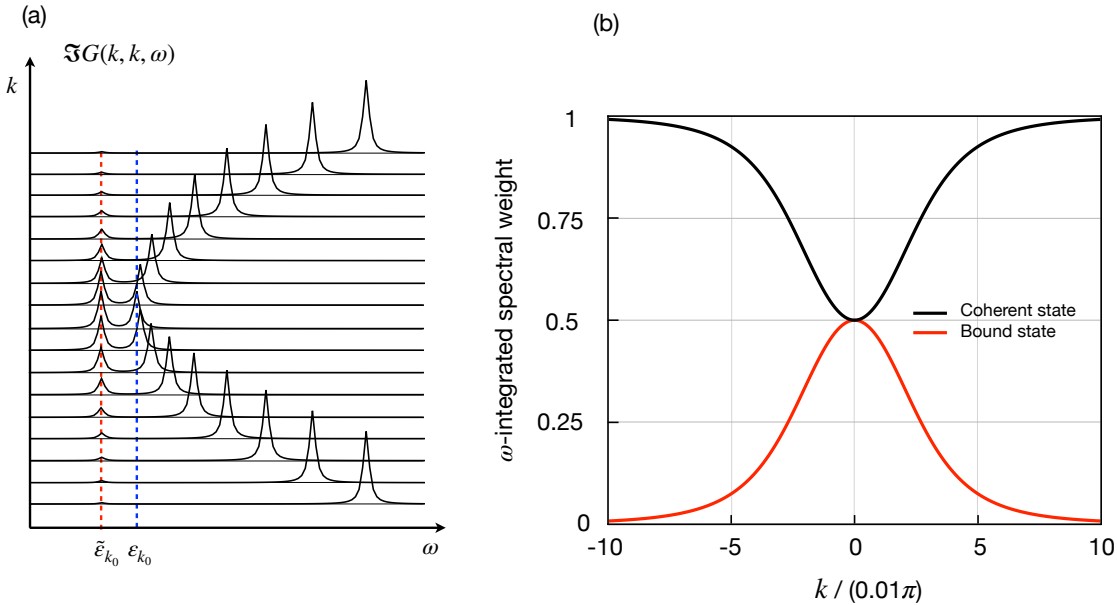

Figure 7: (a) single-particle Green's function for $k \neq k_0$ of a non-magnetic impurity. The level broadening is introduced by hand and the broadening value is fixed to $\eta = 0.1\tilde{\varepsilon}_{k_0}$. (b) Energy-integrated spectral weights of the spectral function according to Eq. (90). The black line shows the coherent excitation and the red line shows the bound state excitation as a function $k$.

Inserting Eq. (89) into the correlation function (84), we can directly evaluate the operator $\delta(\tilde{\mathbf{L}}_V - \omega)$ by simply replacing the transformed Liouville operator $\tilde{\mathbf{L}}_V$ with an eigenvalue of the corresponding single-particle operator it acts on. This is possible because it refers to the diagonalized Hamiltonian $\tilde{\mathcal{H}}_V$. Thus, we obtain for $k \neq k_0$

$$\Im G(k,k,\omega) = \tilde{x}_k^2 \delta(\varepsilon_k - \omega) + \frac{\tilde{y}_k^2}{N^2} \sum_{q(\neq k,k_0)} \left(\frac{\tilde{A}_k + \tilde{A}_q}{\varepsilon_q - \varepsilon_k}\right)^2 \delta(\varepsilon_q - \omega) + \left(\frac{1}{2}\frac{\tilde{\varepsilon}_{k_0} - \varepsilon_{k_0}}{\tilde{\varepsilon}_{k_0} - \varepsilon_k}\right)^2 \delta(\tilde{\varepsilon}_{k_0} - \omega). \tag{90}$$

As in the case of a spin impurity we find the spectral function to exhibit a state at energy $\tilde{\varepsilon}_{k_0}$ *in addition* to the bare energies $\varepsilon_k$ of the free system. The energy $\tilde{\varepsilon}_{k_0}$ of this state is $k$-independent, indicating the state to be local in space – the hallmark of a bound state. Note that the intensity of the bound state (and also the other excitations) depends on $k$. Note that the continuum of states in the second term in Eq. (90) vanishes in the thermodynamic limit due to the factor $1/N^2$. Thus, all excitations exhibit $\delta$-like behavior in the spectral function.

These properties are shown in Fig. 7(a) where the spectral function was calculated using Eq. (90) for a quadratic dispersion $\varepsilon_k$. Furthermore, the spectral weights of the bound state and the dispersive states as calculated from the prefactors in Eq. (90) is shown in Fig. 7(b). We find that the spectral weight of the bound state increases (at the cost of the dispersive states) if it comes closer to the conduction band.

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
