# Peer review of "Renormalization approach to the superconducting Kondo model"

_SciPost Physics, doi:SciPost Phys. 13, 016 (2022)_

## Round 1 · Referee Report · Anonymous (Referee 1) · 2021-11-23

Report

Before proceeding to the report, I'd like to state that my background is single particle theory, including numerical methods. I am therefore knowledgeable in the subject of the manuscript.

In summary, the manuscript proposes a diagonalization procedure for finding bound states in impurity models and applies it to a single impurity in a tight-binding chain, a Shiba state, and a quantum spin impurity in a superconductor.

Novelty of the findings

Finding a bound state spectrum numerically, especially in a noninteracting model, is straightforward to formulate: one chooses a trial energy, computes the bulk Green's function, computes the self-energy of the impurity due to the coupling to the bulk, finds the bound states of the resulting finite Hamiltonian, and iterates this scheme until the trial energy starts coinciding with the bound state energy. A similar approach, together with including mean field corrections would likely also work for the quantum spin example. There are also works, e.g. https://scipost.org/10.21468/SciPostPhys.4.5.026 that consider the numerical aspects like stability or efficiency of the bound state algorithm. Therefore there exist alternative methods to solve this problem. The renormalization approach applied by the authors is reminiscent of a generic construction of the perturbation theory. Upon a closer inspection, however, I realized that the authors formulate an iterative approach to computing a matrix that diagonalizes the Hamiltonian—in other words they seem to just perform iterative numerical diagonalization.

This formulation of the impurity problem is to the best of my knowledge is new, however, given both the familiar nature of the problem, and the relatively standard perturbation expansion approach, the overall idea should be reasonably natural to the experts in the field.

Validity of the reporting

While the overall structure of the manuscript is clear and appears correct, I find that in many aspects the provided information insufficient.

  • The authors refer to a numerical computation that they use to find the bound states, however they do not provide the source code of this computation, and not even an overall description of what they do.
  • The authors use finite sized systems, without addressing the question of how the complexity of the proposed approach depends on the system size. This is especially important because the Green's function-based approach applies directly to infinite systems.
  • The authors' claim of proposing "a (semi)-analytical approach" is misleading: this approach is only analytic as much as is the underlying problem. A generic tight-binding model has no closed form solutions for the bound state problems, and being able to phrase the problem via a system of equations equation doesn't make the approach analytical.

Relevance of the work

Unfortunately, I remain unconvinced about the relevance of the published work: researchers were solving the bound state problem for decades, and in a variety of physical contexts. The authors put forward several claims of usefulness or innovativeness of the manuscript. Below I quote the relevant parts and comment why I find these unconvincing.

we propose a method that merely applies a single unitary transformation.

That is the definition of the eigenvalue decomposition.

This power of our approach is rooted in the fact that bound states manifest themselves as singularities, and that it is therefore sufficient to extract this singular behavior from the full picture to investigate bound state physics.

Existing methods yield the bound state wave function as a linear superposition of exponentially decaying waves. There's no need to read anything out.

our approach directly yields access to momentum-resolved observables

Fourier transform of the bound state wave function yields the same information.

Clarity

Unfortunately, despite the familiar nature of the subject, I found it hard to work my way through the manuscript.

One reason for that is the sequence in which the authors introduce subjects. The authors refer to the material in the following text (for example promising "As we explain below, a convenient Ansatz for $X_V$ is..."), which makes the manuscript logic nonlinear. In this specific example I didn't actually find the promised explanation why this Ansatz is convenient.

Additionally, I can't avoid an impression that the authors introduce complex language to describe something standard. A "single unitary transformation that brings the Hamiltonian to a diagonal form" is the eigenvector decomposition. "Coefficients becoming singular" means that they form a vector in an eigensubspace (by definition). Since these are the key concepts used throughout the manuscript, I am actually not completely certain that the manuscript does anything beyond an iterative finite size diagonalization procedure. In this light, I am especially confused by the statement:

Note that such a treatmant is not a usual diagonalization where the fermion operators in the Hamiltonian are rotated in the operator subspace. In our approach, the one-particle operators are kept in their original basis while the Hamiltonian is renormalized.

A unitary transformation that brings the Hamiltonian to a diagonal form is diagonalization. That the authors choose to call it differently does not change the equations that are solved. A consequence of this description is the introduction of the "bound state momentum" $k_0$, which, as far as I can tell, bears no physical meaning.

A separate point that I did not touch in my discussion above is the quantum spin problem. There, unlike in the other examples, the authors make an approximation (unlabeled equation above Eq. 61), but I am not sure what is its physical meaning. Since the authors eventually obtain an effective single-particle bound state operator, this appears to be some flavor of self-consistent mean field theory, but I cannot tell what it is and when it is valid.

Summary

In view of the above evaluation, I heavily lean towards considering the manuscript unsuitable for publication. In order for me to revisit this evaluation, I would need the authors to implement the changes requested below.

Requested changes

  1. Share the full numerical code used in the manuscript.
  2. Reorganize the manuscript so that it follows a linear logic and does not contain "cliffhangers".
  3. Avoid introducing new concepts where not necessary, in particular in making the seemingly artificial distinction between a diagonalized Hamiltonian and a renormalized Hamiltonian.
  4. Remove the misleading claim of a "semi-analytical" approach.
  5. Explain how the proposed approach is different from an iterative eigenvalue decomposition.
  6. Demonstrate how the results and the computational complexity depend on the number of lattice sites in the chain.

---

## Round 1 · Referee Report · Tomáš Novotný (Referee 2) · 2021-11-28

Strengths

1 - Paper addresses in an innovative way an important problem of ABS states in the superconducting version of the Kondo model.
2 - Solution of this difficult many-body problem is done in a semi-analytical way, which is a remarkable achievement.
3 - The approximative solution found seems to correspond quite well to the numerical solution via NRG, i.e. the solution is meaningful and reliable.

Weaknesses

1- The presentation is in my view not fully optimal, there is too much space and stress given to the semi-trivial preliminaries (impurity scattering and classical spin cases, which can be far more easily solved by other methods such as, e.g., Green's functions, which was correctly pointed out by the first referee), while the really new and important part is relatively succinctly explained. Reverse would be highly welcome. 2 - Connection with numerically exact solution is reduced to a single figure and is not even there done explicitly. Importance of the obtained semi-analytical results would certainly deserve a much better and thorough comparison with the existing numerics.

Report

I consider this paper as a potential breakthrough in the solution of the superconducting version of the Kondo (or Anderson) model. The remarkable achievement of this work is a semianalytical approximate solution of this very complicated many-body problem, which appears to agree quite well with the known numerically exact solution obtained via the numerical renormalization group approach. This strongly suggests that the chosen approximation, although that is by no means clear a priori, captures the most important aspects of the physics of the model. As such I can heartfully recommend this work for publication in SciPost.

However, before that and fully in line with the criticism of the first referee, I strongly recommend the authors to restructure the presentation. The problem with the current version of the manuscript is that it doesn't sufficiently focus on and stress the main achievement which is the solution of the quantum spin problem. Even the title suggests something more general, but that is quite problematic. For single particle problems (the first two parts of the manuscript, i.e., Sec. 2 on nonmagnetic impurity and Sec. 3.1 on classical spin), the presented method is probably one of the most complicated, yet equivalent, approaches to the problem of finding bound states. Here, I must fully agree with probably all the reservations of the first referee. The real achievement though is in the quantum spin part, which to the best of my knowledge gives the first semi-analytical nonperturbative solution to the superconducting Kondo problem. I also understand that the first two parts are preliminaries to this crucial part, but that is likely not too obvious to non-expert (in SC Kondo/Anderson model) readers. I had this impression when reading the paper and the reaction of the first referee fully confirms that.

I would suggest the following: 1. Focus the paper on the quantum spin problem, including the title. 2. Diminish the non-interacting parts; I understand they are necessary technical introductions into the whole formalism, but they are subsidiary, so perhaps move them into appendices or so. 3. Expand the presentation of the quantum spin calculation. It's too short and cannot be easily reproduced. Perhaps also expand explicitly on the concepts taken from references [31] and [37]. 4. Make a better and more thorough comparison of your findings with existing numerically exact results obtained by NRG.

Apart from these general comments I have a couple of specific questions and comments to the text: - Definition of the mean value of an operator between Eqs. (20) and (21) misses the statistical sum/trace over the canonical state in the denominator. - I am wondering why is the k-resolved omega-integrated weight of the bound states in Figs. 2b and 3b exactly 1/2 at the Fermi momentum. You don't show the analogous quantity for the quantum spin, would it be the same? Is it a general/universal property or something specific say for k-independent couplings V? - In Sec. 3.1.1 on pp. 10 and 11 you mention at least twice that the signs in front of the quasiparticle scattering terms should be equal. I understand that it is somehow important for your procedure but I didn't get why it is important.
- 2nd line on p. 15: yield (not yieldS); the end of that paragraph - you write about a TRANSITION between two singlets (BCS and Kondo), that's perhaps a bit dangerous word in that context, since there is no true phase transition unlike between the singlet and doublet but just a smooth crossover. - I am very confused about the statements just below Eq. (71). First of all, in the decoupling you substitute quartic product of quasiparticle operators by a mean value of two of them times the remaining two operators. With respect to which Hamiltonian (or state) is actually the mean value taken? It's not fully clear to me since you make the unitary rotations and things of course correspondingly change... You write that the mean value of number operator alpha_q depends on omega_0. I honestly can't see that but that shouldn't be the sole reason for the necessity of self-consistent solution of Eq. (70) in omega_0 and its ensuing temperature dependence, right? Obviously J_k0 of (71) does depend on omega_0 as it enters the sum in the denominator and the temperature dependence comes in through the mean value even without its explicit omega_0 dependence. So this part is very unclear (and way too short) to me.
- 2nd line on p. 20: you definitely should elaborate on the statement about the correspondence with the former numerical studies [6,20]; the last three lines of that paragraph - I have absolutely NO idea what you mean by the singularity of the J_k coupling, please explain and elaborate - line below Eq. (76): builT (not builD) - Fig. 5, YSR states: why are their weights for various values of J the same (despite their position is moving as it should)? - p. 23, last paragraph of Sec. 3: it might be useful to actually show the behavior of the spectral weight as a function of J

Requested changes

1 - Restructure the manuscript text, suppressing the noninteracting parts and elaborating on the quantum spin problem.
2 - Make a far more thorough comparison of your results for the quantum spin with the existing NRG data.

---

## Round 2 · Referee Report · Tomáš Novotný (Referee 2) · 2022-6-12

Strengths

1 - Novel theoretical approach which is not just purely numerical
2 - Precision of the method comparing well with the numerical benchmark (NRG)

Weaknesses

1 - Rather demanding formalism
2 - Use of somewhat unusual (in the context of SC-Anderson/Kondo model) quantities resolved with respect to the wavevector k; usually local impurity quantities would be primarily addressed

Report

The authors have very carefully addressed both my and other referee's comments and consequently significantly modified the manuscript. I really appreciate that. After careful reading of the updated manuscript (and I must apologize for the long delay) I believe that all my criticism has been adequately dealt with and the new version is suitable for publication.

Requested changes

I have just two minor comments, which should be considered by the authors:

1- There is a sloppy notation in the final part of the manuscript concerning the dimensionless units related to the graphs (see the legend in Fig. 2b, panel headings in Figs. 3 and 4 and again all legends in Fig. 5 as well as the first paragraph in Sec. 3.2 and possibly elsewhere). I believe these should be properly corrected.

2 - I am very confused by the sentence "The YSR states are pinned to the gap edges since the exchange coupling is already relatively large" on top of p. 17. As far as I know, pinning of the YSR states to the gap edge happens for SMALL exchange coupling, larger values on the contrary pull the states more towards the center of the gap. Am I wrong?

---

## Round 2 · Referee Report · Anonymous (Referee 1) · 2022-6-13

Strengths

A clear demonstration of an RG technique that allows to study crossover between different types of ground states.

Weaknesses

Limited clarity of the underlying ideas, limitations, and opportunities for generalization.

Report

Like the other referee, I believe that the updated manuscript is much clearer in explaining the main contribution of the authors and the novelty of the work. I am also satisfied with the way the authors have addressed my concerns.

The manuscript presents a new idea and demonstrates its usefulness, and therefore I think it should be published in SciPost. I also think it may be published in its current form.

Having said that, I would like to also agree with the weaknesses of the approach stated by the other referee: the applicability of the proposed renormalization group procedure seems unclear. After the new version of the manuscript was submitted, the preprint https://doi.org/10.48550/arXiv.2203.13041 was posted. It seems to have a very similar spirit of progressively representing the ground state of an interacting problem using single particle orbitals, but without relying on momentum conservation. Because that preprint appeared later, I don't think the authors need to cite it, however I would appreciate it if the authors comment on the relation between two approaches.

Requested changes

I don't think any changes are necessary, but I think a comment on the relation to https://doi.org/10.48550/arXiv.2203.13041 would be useful to me and to the readers.

---

## Round 2 · Author Response

Dear Dr. Beenakker,

We thank both referees for the careful reading of our manuscript and the helpful comments. Both referees support the publication of our work in SciPost Physics Core if we provide a significantly modified paper in which a convincing restructuring of the presentation of the results is provided. In the resubmitted version, we have carefully addressed this criticism and all other referee’s remarks, as described in the response below. To achieve this goal, we had to revisit the manuscript from its core and also add additional numerical results. We hope that with these extensive modifications and clarification, our manuscript can be accepted for publication.

Yours sincerely, Steffen Sykora and Tobias Meng

RESPONSE TO Anonymous Report 1 on 2021-11-23

"Finding a bound state spectrum numerically, especially in a noninteracting model, is straightforward to formulate: […] A similar approach, together with including mean field corrections would likely also work for the quantum spin example. […] This formulation of the impurity problem is to the best of my knowledge is new, however, given both the familiar nature of the problem, and the relatively standard perturbation expansion approach, the overall idea should be reasonably natural to the experts in the field."

We thank the referee for the appreciation of the novelty of the theoretical approach to the impurity problem considered in our paper. Overall, however, we believe that we did not manage to bring across the main novelty of our approach.

The referee is right in that the solution to non-interacting problems is indeed trivial and standard, and in that sense just „more of the same“. Our focus is, however, on the true quantum spin problem, including its Kondo physics - which in general is a hard problem with rich physics. As one important feedback from the referee’s report, we thus take that the detailed description of non-interacting benchmark cases was blurring the main novel message, which is - again - on quantum spins. Also, as we now detail, our method is more than simple mean field correction, but rather close to flow equations and renormalization approaches.

In the revised manuscript, we have thus moved the description of non-interacting cases to appendices, which focusses the manuscript on its main novel feature. The fact that our approach is tailored towards correlation physics is now highlighted by a more explicit comparison to numerical renormalization group studies reported in the literature, to which our approach compares very well. This highlights that the approach is not just a simple mean-field approach (which for example could not describe Kondo physics faithfully), but significantly goes beyond this. As we describe in more detail now, the approach is philosophically close to flow equation methods.

We would also like to point out that the comments of the second referee lead us to substantially revise the implementation of our renormalization approach, which further strengthens the connection to the flow equations approach.

To iterate, the main novelty reported in this manuscript is the development of an approximation scheme that we use to tackle interaction physics of quantum impurities in superconducting hosts, and that allows to describe quite non-trivial correlation effects.

"The authors refer to a numerical computation that they use to find the bound states, however they do not provide the source code of this computation, and not even an overall description of what they do."

In the new version of the manuscript in the beginning of the new Sec. 3 we have included an extensive description of our numerical computation including starting values, self-consistency loops, and method of the summations. In addition, we have made the source code used for the numerics available for download from the arXiv page of the paper.

The authors use finite sized systems, without addressing the question of how the complexity of the proposed approach depends on the system size. This is especially important because the Green's function-based approach applies directly to infinite systems.

We thank the referee for pointing this out. Since we calculate all summations over momentum states in terms of an energy integral by introducing the BCS density of states, our method also directly applies to infinite systems in the thermodynamic limit. This is possible because for all states except the bound state, the quasiparticle energy remains unchanged. We added a careful explanation of this calculation method to our ‚numerical results‘ section.

"The authors' claim of proposing "a (semi)-analytical approach" is misleading: this approach is only analytic as much as is the underlying problem. A generic tight-binding model has no closed form solutions for the bound state problems, and being able to phrase the problem via a system of equations equation doesn't make the approach analytical."

We thank the referee for this comment. We agree that our original formulation was misleading and could give the impression that we refer to a closed analytic solution (and indeed, there is only a very limited set of problems that in this sense allow for an analytic approach). Instead, our intention was to emphasize that, unlike conventional purely numerical methods, we use an approximation to map the many-body problem of a quantum spin to a solvable effective system of flow equations and self-consistency conditions. Those allow further physical insights to be derived analytically, but cannot be solved analytically in their entirety. We hope that with the new order of the sections where we start with the quantum spin problem, this ambiguity no longer occurs. Nevertheless, we removed the wording ‚semi analytical‘ as the referee stated.

"Unfortunately, I remain unconvinced about the relevance of the published work: researchers were solving the bound state problem for decades, and in a variety of physical contexts. The authors put forward several claims of usefulness or innovativeness of the manuscript. Below I quote the relevant parts and comment why I find these unconvincing."

It was not our intention to present an alternative approach to the classical bound state problem. We understood that the arrangement of the sections in the previous version of the manuscript, where we started with the classical non-magnetic impurity, could indeed have easily led to the above conclusion. Instead, our main intension was to present a new approach to the quantum spin problem in a superconductor (superconducting Kondo model) which can straightforwardly be extended to more impurities. As it is well-known, this problem is highly complex due to the many-body interaction and therefore not trivially accessible by conventional methods. The first two sections of the original version only served to construct the concepts of the method. However, also after reading the report of the second referee, we realised that this way of presentation was misleading. Therefore, to avoid any further misunderstandings, we completely rearranged the sections in the new version of the manuscript. We now explain our complete method including its concepts, renormalization equations, and numerical evaluation in terms of the quantum spin problem and shifted the trivially accessible problems to appendices. We hope that with this reformulation the referee is now convinced that our work is highly relevant.

"Below I quote the relevant parts and comment why I find these unconvincing.

we propose a method that merely applies a single unitary transformation. That is the definition of the eigenvalue decomposition. This power of our approach is rooted in the fact that bound states manifest themselves as singularities, and that it is therefore sufficient to extract this singular behavior from the full picture to investigate bound state physics. Existing methods yield the bound state wave function as a linear superposition of exponentially decaying waves. There's no need to read anything out. our approach directly yields access to momentum-resolved observables Fourier transform of the bound state wave function yields the same information."

As already mentioned above the introduced new concepts appear a bit cumbersome if they are only applied to non-interacting systems where conventional methods work. However, in the Kondo impurity problem, a variant of the scheme presently generalised to superconducting systems was already successful in treating the normal state Kondo lattice. As our main technical advance, the present approach extends these concepts to a superconductor, and explains how the YSR bound states, which appear in addition to a Kondo resonance and are not present in the normal state system, manifest themselves within this approach. We hope that with the mentioned rearrangement of the manuscript it becomes clear that the introduced concepts are really necessary to treat the many-body interactions arising in the quantum spin problem.

"Unfortunately, despite the familiar nature of the subject, I found it hard to work my way through the manuscript. One reason for that is the sequence in which the authors introduce subjects. The authors refer to the material in the following text (for example promising "As we explain below, a convenient Ansatz for XV is..."), which makes the manuscript logic nonlinear. In this specific example I didn't actually find the promised explanation why this Ansatz is convenient."

In the new version of the manuscript all expressions are explained at the same place where they are introduced.

"Additionally, I can't avoid an impression that the authors introduce complex language to describe something standard. A "single unitary transformation that brings the Hamiltonian to a diagonal form" is the eigenvector decomposition. "Coefficients becoming singular" means that they form a vector in an eigensubspace (by definition). Since these are the key concepts used throughout the manuscript, I am actually not completely certain that the manuscript does anything beyond an iterative finite size diagonalization procedure. In this light, I am especially confused by the statement:

Note that such a treatmant is not a usual diagonalization where the fermion operators in the Hamiltonian are rotated in the operator subspace. In our approach, the one-particle operators are kept in their original basis while the Hamiltonian is renormalized. A unitary transformation that brings the Hamiltonian to a diagonal form is diagonalization. That the authors choose to call it differently does not change the equations that are solved. A consequence of this description is the introduction of the "bound state momentum“ k0, which, as far as I can tell, bears no physical meaning."

We thank the referee for all these comments, and we also realized that these formulations were not fully optimal. In fact, we do a diagonalization of the Hamiltonian. We have removed the related wording in the new version and explain in much more detail the procedure of our unitary transformation.

"A separate point that I did not touch in my discussion above is the quantum spin problem. There, unlike in the other examples, the authors make an approximation (unlabeled equation above Eq. 61), but I am not sure what is its physical meaning. Since the authors eventually obtain an effective single-particle bound state operator, this appears to be some flavor of self-consistent mean field theory, but I cannot tell what it is and when it is valid."

As explained above, this part is actually the main result of our work, and we sincerely apologise that our former presentation blurred this aspect. We have thus substantially revised the manuscript.

Concerning the question of validity, we believe that the validity of our approximation scheme can be gauged by the extensive comparison to other methods (NRG in particular), showing very similar results. In addition, the approximation scheme was used successfully in previous works in the normal-state Kondo lattice model. Furthermore, it should be mentioned that this scheme is commonly used in other diagonalization schemes of Hamiltonians as for example the flow equation method by Wegner/Kehrein. Nevertheless, in our new Sec. 2.1 we extended the discussion on this aspect and explained in much more detail which consequences one would expect.

RESPONSE TO Report 2 by Tomáš Novotný on 2021-11-28

"I consider this paper as a potential breakthrough in the solution of the superconducting version of the Kondo (or Anderson) model. The remarkable achievement of this work is a semianalytical approximate solution of this very complicated many-body problem, which appears to agree quite well with the known numerically exact solution obtained via the numerical renormalization group approach. This strongly suggests that the chosen approximation, although that is by no means clear a priori, captures the most important aspects of the physics of the model. As such I can heartfully recommend this work for publication in SciPost."

We thank the referee for the overall appreciation of our work.

"However, before that and fully in line with the criticism of the first referee, I strongly recommend the authors to restructure the presentation. The problem with the current version of the manuscript is that it doesn't sufficiently focus on and stress the main achievement which is the solution of the quantum spin problem. Even the title suggests something more general, but that is quite problematic. For single particle problems (the first two parts of the manuscript, i.e., Sec. 2 on nonmagnetic impurity and Sec. 3.1 on classical spin), the presented method is probably one of the most complicated, yet equivalent, approaches to the problem of finding bound states. Here, I must fully agree with probably all the reservations of the first referee. The real achievement though is in the quantum spin part, which to the best of my knowledge gives the first semi-analytical nonperturbative solution to the superconducting Kondo problem. I also understand that the first two parts are preliminaries to this crucial part, but that is likely not too obvious to non-expert (in SC Kondo/Anderson model) readers. I had this impression when reading the paper and the reaction of the first referee fully confirms that."

We thank the referee for pointing this out. We fully agree with all the comments and follow all the suggestions. We have completely restructured the manuscript and our method is now constructed in terms of the quantum spin problem. For details see our response to the specific comments below.

"Focus the paper on the quantum spin problem, including the title."

Following this suggestion the new version of the manuscript now focuses on the quantum spin problem. Now we construct our renormalization approach fully in terms of this problem and start already in Sec. 2 after the introduction section by introducing the model, setting up all equations including the evaluation of expectation values. The new Sec. 3 shows corresponding numerical results. The title is now changed to ‚Renormalization approach to the superconducting Kondo model‘.

"2. Diminish the non-interacting parts; I understand they are necessary technical introductions into the whole formalism, but they are subsidiary, so perhaps move them into appendices or so."

We have shifted the non-interacting parts, i.e. the classical spin case and the non-magnetic impurity, to the appendices A and B. The whole formalism is now built in terms of the interacting problem.

"3. Expand the presentation of the quantum spin calculation. It's too short and cannot be easily reproduced. Perhaps also expand explicitly on the concepts taken from references [31] and [37]."

In the new version we have significantly extended the presentation of our numerical results. In the new Sec. 3 we have additionally included results for the Kondo resonance and the influence of temperature and spin on the YSR states. All the results are carefully compared with former numerical studies.

In addition, the referee’s comments prompted us to test different variants of our approximation scheme. We found that when changing the scheme from two steps to a scheme with many smaller steps, our method can be cast into a form closer to the familiar flow equation approach (which we think adds to the understanding of the approach itself), and also yields better results as compared to NRG data. Instead of applying a two-step procedure for the unitary transformation as described in the old version, we have therefore now updated our diagonalization scheme in closer connection to Ref. [31] which is based on a multiple step procedure. The way of finding the corresponding generator of the transformation should now become more clear.

"4. Make a better and more thorough comparison of your findings with existing numerically exact results obtained by NRG."

Following this suggestion we have extended the presentation of our numerical results significantly. At first we have added a panel (b) to the YSR energy plot in the new Fig. 2 where we show the calculated density of states at larger values of the Kondo temperature in the strong coupling regime. A strong enhancement of the spectral intensity around the normal state Fermi level is found which marks the crossover into the Kondo resonance regime. In addition, the inset shows the spectral weight of the YSR bound state around the transition to the Kondo regime. In agreement with exact numerical studies a jump-like behaviour is found. This jump is one of the examples where the new scheme with many steps is better than the old scheme (and we could track down the point in the approximations at which the jump disappeared in the old scheme, see below). Then, in the new Fig. 5 the influence of spin and temperature to the YSR bound state energy and spectral intensity is shown. The variation of the YSR state energy with temperature as shown in panel (a) agrees very well with a former NRG study [6]. This is even better seen in panels (b,d) where the momentum integrated spectral function shows the pair of YSR bound states in a narrow energy region inside the superconducting gap. This has also explicitly been calculated in Ref. [6]. Shown are the spectra at two temperatures (blue and red line) and two different Kondo temperatures (upper/lower panel). As is seen a very good agreement with Ref. [6] is found.

"Definition of the mean value of an operator between Eqs. (20) and (21) misses the statistical sum/trace over the canonical state in the denominator."

We thank the referee for this comment. In the new Eqs. (40-42) the statistical sum is now included.

"I am wondering why is the k-resolved omega-integrated weight of the bound states in Figs. 2b and 3b exactly 1/2 at the Fermi momentum. You don't show the analogous quantity for the quantum spin, would it be the same? Is it a general/universal property or something specific say for k-independent couplings V?"

Indeed, the spectral weight of the bound state is exactly 1/2 at the Fermi momentum. However, this only applies to the non-interacting systems. This property can be recognized, for example, by the equation (89), which describes the distribution of weight among the individual parts of the one-particle operator in a trivial scattering scenario. The last term, which is the creation of the bound state, takes the value 1/2 if k=k_0. This is a consequence of the unitary transformation of the one-particle operators which is equally carried out also for the Hamiltonian leading to the renormalization given by Eq. (78). Therefore, this feature is rather universal for non-interacting systems and not dependent on the shape of V. However, in the interacting case this property is not present in general. Since the bound state energy is a solution of Eq. (35), where expectation values enter, there is no universal spectral weight of the bound state as is seen in the last term of Eq. (48). This can be also seen in the numerical solution from a comparison of the bound state peaks in Fig. 4(c), where in the strong coupling case the intensity of the bound state is much higher in comparison with the coherent peaks.

"In Sec. 3.1.1 on pp. 10 and 11 you mention at least twice that the signs in front of the quasiparticle scattering terms should be equal. I understand that it is somehow important for your procedure but I didn't get why it is important. "

While restructuring our manuscript we realized that in the original formulation of the two step procedure of the unitary transformation this property has been unjustifiably assigned to a physical meaning. The change of the sign during the original first transformation step was purely technical. In the new formulation of the stepwise unitary transformation all necessary terms are in one uniform generator as given by Eq. (12). With the new choice of the generator, no sign change appears anymore.

"2nd line on p. 15: yield (not yieldS); the end of that paragraph - you write about a TRANSITION between two singlets (BCS and Kondo), that's perhaps a bit dangerous word in that context, since there is no true phase transition unlike between the singlet and doublet but just a smooth crossover."

We thank the referee for these comments. We fully agree that the described change of the scenarios does not correspond to a true phase transition. Therefore, we replaced ‚transition‘ with ‚crossover‘.

"I am very confused about the statements just below Eq. (71). First of all, in the decoupling you substitute quartic product of quasiparticle operators by a mean value of two of them times the remaining two operators. With respect to which Hamiltonian (or state) is actually the mean value taken? It's not fully clear to me since you make the unitary rotations and things of course correspondingly change... "

In the new version of the manuscript, we extensively describe in the new Sec. 2.3 how expectation values are taken and defined. See in particular the Eqs. (25) and (39-41). As the referee states the Hamiltonian with which the expectation value is taken should change with the unitary transformation. This is exactly what we do and actually it is expressed in the dependence of the expectation values on our renormalization parameter lambda.

"You write that the mean value of number operator alpha_q depends on omega_0. I honestly can't see that but that shouldn't be the sole reason for the necessity of self-consistent solution of Eq. (70) in omega_0 and its ensuing temperature dependence, right? Obviously J_k0 of (71) does depend on omega_0 as it enters the sum in the denominator and the temperature dependence comes in through the mean value even without its explicit omega_0 dependence. So this part is very unclear (and way too short) to me."

We have significantly extended the formalism leading to the bound state energy. It is determined by the self-consistent solution of Eq. (35), where also expectation values enter. Therefore, in the quantum spin case, temperature enters and influences the bound state energy (via Eq. (35)) and also its quasiparticle weight through Eqs. (48) and (37). We hope that with the new formulation of the unitary transformation in the new version of the manuscript the effect of the temperature becomes more clear.

"2nd line on p. 20: you definitely should elaborate on the statement about the correspondence with the former numerical studies [6,20]; the last three lines of that paragraph - I have absolutely NO idea what you mean by the singularity of the J_k coupling, please explain and elaborate"

As already stated above we have extended our comparison to the former numerical studies. We now present further results, in particular quasiparticle weight of the bound state and its temperature dependence, that clearly show the very good agreement with the mentioned studies. Furthermore, within our new formalism it should become clear that the mentioned singularity of J_k is assigned to the YSR bound state. However, in the strong coupling case there are additional diverging contributions to J_k in the momentum range around the Fermi level leading to the appearance of the Kondo resonance. These excitations are placed in the spectral function outside the superconducting gap and they scale with the Kondo temperature. This difference is now described in much more detail, in particular in the context of Fig. 2(b).

"line below Eq. (76): builT (not builD)"

We have removed this paragraph since it doesn't apply anymore in the new formulation.

"Fig. 5, YSR states: why are their weights for various values of J the same (despite their position is moving as it should)?"

In the old version we realized that in the renormalized quasiparticle operator (Eq. (73) in the old version), several terms were missing and not included in the calculations. This concerns a momentum summation of all momentum vectors different from the bound state. In the updated variant of our approximation scheme using the new Eq. (48), they appear as the second term (with momentum summation). These particular contributions lead to an additional spectral weight transfer away from the bound state which depends on J. As is seen in the new Fig. 3, the weights now do change as the referee correctly expected.

"p. 23, last paragraph of Sec. 3: it might be useful to actually show the behavior of the spectral weight as a function of J"

In the new version this behavior is explicitly shown in the inset of the new Fig. 2(b).

---

## Round 2 · List of Changes

1. We have changed the title to 'Renormalization approach to the superconducting Kondo model'
2. We have adjusted the abstract to the new structure of the manuscript.
3. Extensive restructuring of the manuscript. New starting point is the quantum spin problem. Non-interacting problems were shifted to the appendix.
4. New scheme of the unitary transformation which is closer to the flow equation approaches.
5. Adjustment of all figures according to the new calculation scheme.
6. Extension of the numerical results. New panel (b) of Fig. 2. New Fig. 5 with effect of temperature and spin.
7. New discussion of the Kondo resonance in terms of additional contributions to the quasiparticle operator.
8. Addition of the Reference Phys. Rev. B 94, 085151 (2016).

---

## Editorial Decision

published